# MARKET GAMES FOR GENERATIVE MODELS: EQUILIBRIA, WELFARE, AND STRATEGIC ENTRY

**Xiukun Wei, Min Shi, Xueru Zhang** *
The Ohio State University
{wei.1418,shi.1532,zhang.12807}@osu.edu

## ABSTRACT

Generative model ecosystems increasingly operate as competitive multi-platform markets, where platforms strategically select models from a shared pool and users with heterogeneous preferences choose among them. Understanding how platforms interact, when market equilibria exist, how outcomes are shaped by model-providers, platforms, and user behavior, and how social welfare is affected is critical for fostering a beneficial market environment. In this paper, we formalize a three-layer *model-platform-user* market game and identify conditions for the existence of pure Nash equilibrium. Our analysis shows that market structure, whether platforms converge on similar models or differentiate by selecting distinct ones, depends not only on models' global average performance but also on their localized attraction to user groups. We further examine welfare outcomes and show that expanding the model pool does not necessarily increase user welfare or market diversity. Finally, we design novel best-response training schemes that allow model providers to strategically introduce new models into competitive markets.

## 1 INTRODUCTION

Generative models are no longer developed in isolation. They now operate within competitive, multi-platform markets, where platforms strategically deploy models to attract heterogeneous user groups and compete for market share. For example, Microsoft Azure and Amazon Bedrock compete to license foundation models to enterprises (Buchanan, 2024; Janakiram, 2023), Canva and Adobe Firefly compete to attract designers by integrating state-of-the-art generative models (Newsroom, 2024; Weatherbed, 2024), and platforms like Midjourney and Stability AI compete directly for end users seeking creative tools (Staff, 2025a;b). Understanding the behavior of such markets is crucial for guiding governance, policy, and the design of trustworthy AI ecosystems.

Prior work has largely focused on a two-layer market, where model developers also operate the delivery platforms and offer services directly to users (Einav & Rosenfeld, 2025; Jagadeesan et al., 2023b; Raghavan, 2024; Taitler & Ben-Porat, 2025). Within this setting, a substantial body of literature examines discriminative scenarios in which (binary) classifiers are trained and used by end users. For instance, Einav & Rosenfeld (2025) show that platforms often prioritize capturing user share over maximizing predictive accuracy, leading to equilibrium states that deviate from socially optimal outcomes. Similarly, Jagadeesan et al. (2023b) demonstrate that even when individual classifiers achieve lower Bayes risk, competition can perversely reduce overall social welfare. By contrast, the literature on generative model markets remains sparse, with only a few recent studies analyzing how competition shapes overall welfare outcomes (Taitler & Ben-Porat, 2025; Raghavan, 2024). Specifically, Taitler & Ben-Porat (2025) identify a counterintuitive phenomenon: adding more models can paradoxically decrease user welfare. More recently, Raghavan (2024) find that competitive pressures often reduce diversity, though stronger competition can partially mitigate homogenization, and that models performing well in isolation may fail in competitive environments. Collectively, these studies highlight that, in generative markets, improvements in individual model performance do not necessarily translate into greater welfare or diversity.

However, the generative ecosystem is increasingly structured as a three-layer market (Fallah et al., 2024): model providers develop models and license them to platforms, which in turn deliver services

---

*Corresponding author.

to end users. Unlike the two-layer setting, where model developers both build and operate the delivery platforms, in the three-layer market, platforms act as intermediaries: they decide which models to adopt and deploy, ultimately shaping how users experience generative AI. For instance, Azure OpenAI Service (Azure, 2025) supplies GPT-family models through Microsoft Azure, enabling enterprise clients to embed them into a wide range of applications; Cohere (Cohere, 2025) provides large language models as APIs for enterprises, serving platforms rather than end users directly; and Canva (Canva, 2025) integrates external models such as Stable Diffusion and Leonardo Phoenix into its design suite, allowing millions of users to access generative capabilities without ever selecting the underlying model themselves. In all these cases, platforms are the direct consumers of models, while users experience only the models that platforms choose.

In this work, we formalize the market as a three-layer **model-platform-user** game, in which heterogeneous users choose the platform that best aligns with their preferences, while platforms strategically adopt the models from providers that maximize their market share (Fig. 1). We then conduct a rigorous analysis of how platform-level competition influences user welfare, diversity, and equilibrium outcomes. Our main contributions and findings are summarized below:

1. In Section 2, we formalize the model-platform-user game and show that when users make hard selections on platforms, the resulting game among platforms may not admit pure Nash equilibria.

2. In Section 3, we identify conditions for the existence of pure Nash equilibria. Crucially, we analyze market structure at equilibrium and derive conditions for both fully differentiated equilibria (all platforms choose distinct models) and homogeneous equilibria (all platforms converge on the same model). We show that market structure is determined not only by models' average performance but also by their deviation advantage to heterogeneous users.

3. In Section 4, we analyze market diversity and user welfare. We show that equilibrium may not achieve the socially optimal outcome that maximizes user welfare, and that increasing competition (e.g., adding more platforms or models) does not necessarily improve user welfare or market diversity. This finding aligns with recent observations of growing homogenization in generative model markets (Zhang et al., 2025; Wu et al., 2025).

4. In Section 6, we take the model providers' perspective and design best-response training schemes that allow a provider to introduce a new model effectively into the competitive market.

5. In Section 7, we conduct experiments on both synthetic and real data to validate our theorems.

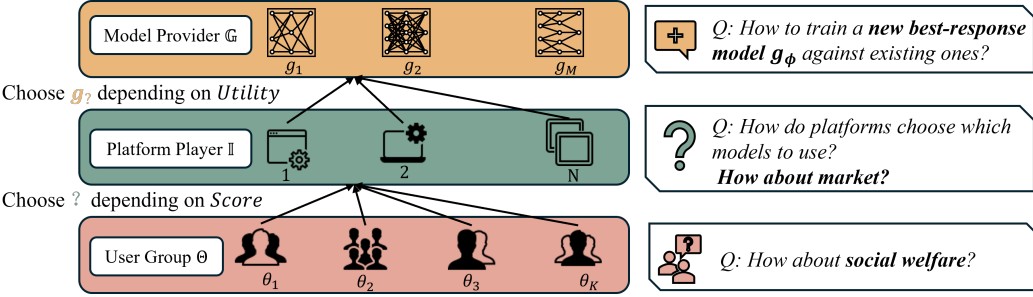

Figure 1: The three-layer model-platform-user market structure. Model providers develop generative models, platforms select models to deploy, and heterogeneous users choose platforms.

## 2 PROBLEM MODEL

Consider a three-layer generative model service market as shown in Fig. 1, which consists of:

- **Model Layer**: A set of generative models $\mathbb{G} = \{g_1, \ldots, g_M\}$ trained by $M$ model providers. Each model $g_i$, $i \in \mathbb{M} = \{1, \ldots, M\}$ is characterized as a data distribution over domain $\mathcal{X}$.

- **Platform Layer**: A set of platform-players $\mathbb{I} = \{1, \ldots, N\}$. From the provider, each platform $i$ selects a model $f_i \in \mathbb{M}$ to serve its users. Denote the platform selection as $\boldsymbol{f} = (f_1, \ldots, f_N)$.

- **User Layer**: A population of heterogeneous users categorized into types $\Theta = \{\boldsymbol{\theta}_1, \ldots, \boldsymbol{\theta}_K\} \subseteq \mathbb{R}^d$ with distribution $\{\pi_{\boldsymbol{\theta}}\}_{\boldsymbol{\theta} \in \Theta}$ such that $\sum_{\boldsymbol{\theta} \in \Theta} \pi_{\boldsymbol{\theta}} = 1$. For each user type $\boldsymbol{\theta}_k$, let $r_{\boldsymbol{\theta}_k}(x)$ be the underlying reward function indicating their preference over generated content $x \in \mathcal{X}$.

**User's choice of platform.** Given platform selections, users of type $\boldsymbol{\theta}$ choose platforms to interact based on the generation ability of selected models. Let $S_j(\boldsymbol{\theta}) := \mathbb{E}_{x \sim g_j}[r_{\boldsymbol{\theta}}(x)]$ be *score* measuring the quality of contents generated by model $g_j$ for user type $\boldsymbol{\theta}$; higher $S_j(\boldsymbol{\theta})$ indicates better alignment with user preferences. Then, the probability for users of type $\boldsymbol{\theta}$ selecting platform $i$ is given by:

$$p_i(\boldsymbol{\theta}) := \begin{cases} 0 & \text{if } f_i \notin \arg\max_{i' \in [N]} S_{f_{i'}}(\boldsymbol{\theta}) \\ \frac{1}{\left|\arg\max_{i' \in [N]} S_{f_{i'}}(\boldsymbol{\theta})\right|} & \text{if } f_i \in \arg\max_{i' \in [N]} S_{f_{i'}}(\boldsymbol{\theta}) \end{cases} \tag{1}$$

That is, users select the platform with the highest score, breaking ties uniformly at random. This hardmax choice model is a standard simplification commonly used in prior work on platform or model selection (Jagadeesan et al., 2023b;a; Mansour et al., 2018). In Section 5, we discuss how our analysis and results can be extended to the softmax user choice model.

**Platform's choice of models.** Each platform $i \in [N]$ strategically selects a model $f_i^*$ from the model-provider to compete for market share, in response to other platforms' selections:

$$f_i^* = \arg\max_{f_i \in \mathbb{M}} U_i(f_i; \boldsymbol{f}_{-i}^*) \tag{2}$$

where $U_i(f_i; \boldsymbol{f}_{-i}) := \sum_{\boldsymbol{\theta} \in \Theta} \pi_{\boldsymbol{\theta}} \cdot p_i(\boldsymbol{\theta}) \cdot S_{f_i}(\boldsymbol{\theta})$ is the utility function capturing the market share of platform. Denote profile $\boldsymbol{f} = (f_1, \cdots, f_N)$ as the strategies of all platforms, and $\boldsymbol{f}_{-i}$ strategies of all excluding the platform $i$. We consider a normal-form game $\mathcal{G}(\mathbb{G}, \mathbb{I}, \Theta)$ in which each platform chooses a model to maximize its utility.

**Definition 2.1** (Nash Equilibrium). We say a strategy profile $\boldsymbol{f}^* = (f_1^*, \cdots, f_N^*)$ is a *pure Nash equilibrium* (PNE) if no platform can improve its utility by unilaterally deviating, i.e., $U_i(f_i^*; \boldsymbol{f}_{-i}^*) \geq U_i(f_i; \boldsymbol{f}_{-i}^*), \forall i$. A pure Nash equilibrium is **fully differentiated equilibrium** if all platforms choose distinct models: $|\{f_1^*, \cdots, f_N^*\}| = N$. A pure Nash equilibrium is **homogeneous equilibrium** if all platforms choose the same model: $|\{f_1^*, \cdots, f_N^*\}| = 1$.

By Nash's classical theorem, every finite game admits at least one *mixed-strategy* Nash equilibrium, where each player $i$ randomizes over actions according to a probability distribution (Monderer & Shapley, 1996). However, when users make deterministic selections as in Eq. 1, we show in Proposition 2.2 that a pure Nash equilibrium may not exist.

**Proposition 2.2.** *[Nonexistence of PNE] Consider the game $\mathcal{G}(\mathbb{G}, \mathbb{I}, \Theta)$ with finite sets of platforms $\mathbb{I}$, models $\mathbb{G}$, and user types $\Theta$, where each platform $i$ chooses a model $f_i \in \mathbb{M}$ based on Eq. 1. The game may not admit a pure-strategy Nash equilibrium $\boldsymbol{f}^*$.*

To prove Proposition 2.2, we provide a counterexample in Appendix C.1. The existence of a pure Nash equilibrium can be examined via *best-response dynamics*: starting from any profile $\boldsymbol{f}$, platforms sequentially update their strategies as best responses to the current strategies of others. When a pure equilibrium does not exist, this process fails to converge and instead enters cycles.

**Definition 2.3** (Best-Response Cycle). A *best-response cycle* is a finite sequence of strategy profiles $\boldsymbol{f}^{(1)}, \boldsymbol{f}^{(2)}, \ldots, \boldsymbol{f}^{(L)}$ in best-response dynamics such that: (i) at each step $t$, only one platform changes its strategy, and it does so as a best response to $\boldsymbol{f}^{(t)}$, and (ii) the sequence eventually returns to the initial profile, i.e., $\boldsymbol{f}^{(L+1)} = \boldsymbol{f}^{(1)}$.

**User welfare & market diversity.** Next, we introduce metrics we will use for analyzing the market.

**Definition 2.4** (Coverage Value). For a strategy profile $\boldsymbol{f} = (f_1, \ldots, f_N)$ with $f_i \in \mathbb{M}$. define the *coverage value* of $\boldsymbol{f}$ as $V(\boldsymbol{f}) := \sum_{\boldsymbol{\theta} \in \Theta} \pi_{\boldsymbol{\theta}} \max_{1 \leq i \leq N} S_{f_i}(\boldsymbol{\theta})$, i.e., the expected quality users receive from their best available platforms.

**Definition 2.5** (User Welfare). Define *user welfare* $W$ as $W := V(\boldsymbol{f}^*)$ if the game $\mathcal{G}(\mathbb{G}, \mathbb{I}, \Theta)$ admits a pure Nash equilibrium $\boldsymbol{f}^*$; and $W := \frac{1}{L} \sum_{t=1}^{L} V(\boldsymbol{f}^{(t)})$ if it enters best-response cycle.

**Definition 2.6** (Social Optimum Welfare). The *social optimum welfare* is defined as the highest coverage value achievable by any strategy profile. That is, $W_{\text{opt}} := \max_{\boldsymbol{f} \in \mathbb{M}^N} V(\boldsymbol{f})$.

**Definition 2.7** (Herfindahl-Hirschman Index (HHI) Diversity). For strategy $\boldsymbol{f} = (f_1, \ldots, f_N)$ with $f_i \in \mathbb{M}$, let $(\mu_1, \mu_2, \ldots, \mu_N)$ be the *market share vector* of platforms, where $\mu_i = \sum_{\boldsymbol{\theta} \in \Theta} \pi_{\boldsymbol{\theta}} p_i(\boldsymbol{\theta})$. *HHI diversity* is defined as $D_{\text{HHI}}(\boldsymbol{f}) := \sum_{i=1}^{N} \mu_i^2$.

HHI diversity measures how evenly users are distributed across platforms. Its value lies in $[\frac{1}{N}, 1]$, taking value $\frac{1}{N}$ when users are evenly split across all platforms (maximum diversity) and 1 when all users concentrate on a single platform (no diversity).

**Definition 2.8** (Support Diversity). For strategy $\boldsymbol{f} = (f_1, \ldots, f_N)$ with $f_i \in \mathbb{M}$, the *support diversity* is defined as $D_{\mathrm{supp}}(\boldsymbol{f}) := |\{m \in \mathbb{M} \mid \exists i \in \mathbb{I}, f_i = m\}|$

Support diversity measures the number of distinct models adopted by platforms, taking integer values between 1 and $N$. Larger values indicate that more models are represented in the market, reflecting greater model diversity.

**Objectives.** With the platform selection game and evaluation metrics in place, the remainder of this paper aims to address the following key questions about competitive generative model markets:

- Under what conditions do pure Nash equilibria (PNE) arise, and when do platforms converge to differentiated versus homogeneous structures?

- How does the user welfare depend on the number of platforms and models, and how does it compare to the social optimum?

- From the model-provider's perspective, how to design new generative models that can successfully enter the market and be strategically adopted by competing platforms?

## 3 EQUILIBRIUM ANALYSIS AND MARKET STRUCTURE

Proposition 2.2 showed that a pure Nash equilibrium (PNE) may not exist in the platform selection game. We now investigate the conditions under which a PNE does exist. To facilitate the analysis, we first introduce some basic notations.

**Definition 3.1** (Average Score). Let *average score* of model $g_j$ be defined as $T_j := \sum_{\boldsymbol{\theta} \in \Theta} \pi_{\boldsymbol{\theta}} \cdot S_j(\boldsymbol{\theta})$

**Definition 3.2** (Attraction Term and Deviation Advantage). For a strategy profile $\boldsymbol{f} = (f_1, \ldots, f_N)$ with $f_i \in \mathbb{M}$, define $\mathbb{A}_{\boldsymbol{f}}(\boldsymbol{\theta}) := \arg\max_{1 \leq i \leq N} S_{f_i}(\boldsymbol{\theta})$ as the set of maximizers for a user type $\boldsymbol{\theta}$, and let $A_{\boldsymbol{f}}(\boldsymbol{\theta}) := |\mathbb{A}_{\boldsymbol{f}}(\boldsymbol{\theta})|$ be the number of platforms tied for the maximum. Then, the *attraction term* for $f_i$ in strategy $\boldsymbol{f}$ is defined as

$$Z_{f_i}(\boldsymbol{\theta}; \boldsymbol{f}) := \begin{cases} \dfrac{N - A_{\boldsymbol{f}}(\boldsymbol{\theta})}{A_{\boldsymbol{f}}(\boldsymbol{\theta})} S_{f_i}(\boldsymbol{\theta}) & \text{if } f_i \in \mathbb{A}_{\boldsymbol{f}}(\boldsymbol{\theta}) \\ -S_{f_i}(\boldsymbol{\theta}), & \text{otherwise} \end{cases} \tag{3}$$

The *deviation advantage* for $f_i$ under strategy $\boldsymbol{f}$ is defined as

$$\delta_{f_i}(\boldsymbol{f}) := \sum_{\boldsymbol{\theta} \in \Theta} \pi_{\boldsymbol{\theta}} \cdot Z_{f_i}(\boldsymbol{\theta}; \boldsymbol{f}) \tag{4}$$

Intuitively, the attraction term $Z_{f_i}(\boldsymbol{\theta}; \boldsymbol{f})$ quantifies how much $f_i$ benefits when it is among the winners for type $\boldsymbol{\theta}$, and how much it loses otherwise. Aggregated across all user types, the deviation advantage $\delta_{f_i}(\boldsymbol{f})$ represents the net gain of $f_i$ relative to its competitors under the current strategy.

**Proposition 3.3.** *[Utility Decomposition.] The expected utility $U_i(f_i; \boldsymbol{f}_{-i})$ of platform $i$ in Eq. 2 can be decomposed into $T_{f_i}$ and $\delta_{f_i}(\boldsymbol{f})$ as:*

$$U_i(f_i; \boldsymbol{f}_{-i}) = \frac{1}{N}\left(T_{f_i} + \delta_{f_i}(\boldsymbol{f})\right). \tag{5}$$

**Lemma 3.4.** *[Existence of Equilibrium] Consider the game $\mathcal{G}(\mathbb{G}, \mathbb{I}, \Theta)$ with finite user types $\Theta$, and $N$ platforms choosing from $M$ models $\mathbb{G}$, where $M \geq N$. A **fully differentiated equilibrium** $\boldsymbol{f}^* = (f_1^*, \ldots, f_N^*)$ exists if and only if for every platform $i$ and every alternative model $f_i \in \mathbb{G} \setminus \{f_i^*\}$,*

$$T_{f_i^*} - T_{f_i} \geq \delta_{f_i}(\boldsymbol{f}_{-i}^* \cup f_i) - \delta_{f_i^*}(\boldsymbol{f}^*) \tag{6}$$

*A **homogeneous equilibrium** $\boldsymbol{f}^* = (f_1^*, \ldots, f_N^*)$, $f_i^* = m$ exists if and only if for some $m \in \mathbb{M}$,*

$$T_m - T_{f_i} \geq \delta_{f_i}(\boldsymbol{f}_{-m}^* \cup f_i) - \delta_m(\boldsymbol{f}^*) \tag{7}$$

**Scenario A:** fully differentiated equilibrium

$$\pi_{\boldsymbol{\theta}_A} = \pi_{\boldsymbol{\theta}_B} = 0.5$$

|              | $S_1(\boldsymbol{\theta})$ | $S_2(\boldsymbol{\theta})$ |
| ------------ | -------------------------- | -------------------------- |
| $\boldsymbol{\theta}_A$ | 0.90 | 0.85 |
| $\boldsymbol{\theta}_B$ | 0.35 | 0.80 |

$$\boldsymbol{f}^* = (g_1, g_2)$$

**Scenario B:** homogeneous equilibrium

$$\pi_{\boldsymbol{\theta}_A} = \pi_{\boldsymbol{\theta}_B} = 0.5$$

|              | $S_1(\boldsymbol{\theta})$ | $S_2(\boldsymbol{\theta})$ |
| ------------ | -------------------------- | -------------------------- |
| $\boldsymbol{\theta}_A$ | 0.60 | 0.70 |
| $\boldsymbol{\theta}_B$ | 0.65 | 0.95 |

$$\boldsymbol{f}^* = (g_2, g_2)$$

Figure 2: Two scenarios with the same average score gap $|T_2 - T_1| = 0.20$ but different deviation advantage $(\delta_1, \delta_2)$ for $N = 2$ and $M = 2$, resulting in opposite equilibrium outcomes. In Scenario A, although $g_1$ has a lower average score ($T_1 < T_2$), it is still chosen in equilibrium because its strong advantage on the high-weight type $\boldsymbol{\theta}_A$ satisfies the differentiation condition. Removing this type-specific advantage in Scenario B breaks the condition, leading to a homogeneous equilibrium on $g_2$. These scenarios demonstrate that market structure is not determined solely by average performance; a strong local advantage in high-weight user segments can sustain a model's presence in equilibrium. Full calculation details are provided in Section C.4.

The proof is given in Section C.3. Lemma 3.4 shows that market structure is determined by the balance between average performance $T$ and the deviation advantage $\delta$. When average performance is uniformly strong across models, platforms tend to converge on the single best model, where performance alone cannot sustain multi-model entry. In contrast, when models differ in whom they serve best, even uniformly weak models can secure adoption as platforms specialize, yielding a differentiated market. To illustrate this, we provide an example in Fig. 2.

Lemma 3.4 also implies that market structure depends on the user distribution. Building on this, Corollary 3.5 shows that when the user distribution is centralized, i.e., a single user type constitutes a large fraction of the population, the market tends to converge to a homogeneous structure.

**Corollary 3.5** (High User Centralization $\Rightarrow$ Homogeneous Equilibrium). *Assume there exists a dominant user type $\boldsymbol{\theta}^\star$ with fraction $\pi_{\boldsymbol{\theta}}^\star$ and a model $m$ satisfying: $\forall j \neq m$, $S_m(\boldsymbol{\theta}^\star) - S_j(\boldsymbol{\theta}^\star) \geq \rho > 0$ and $\forall \boldsymbol{\theta} \neq \boldsymbol{\theta}^\star, j \neq m$, $|S_j(\boldsymbol{\theta}) - S_m(\boldsymbol{\theta})| \leq \Gamma$. If $\pi_{\boldsymbol{\theta}}^\star$ is sufficiently large and satisfies*

$$\pi_{\boldsymbol{\theta}}^\star \geq 1 - \frac{1}{1 + 2\frac{\Gamma}{\rho}}$$

*then the homogeneous strategy $\boldsymbol{f}^* = (m, \ldots, m)$ is a pure-strategy Nash equilibrium.*

Intuitively, $\rho$ measures strength of the "majority advantage" of model, $\Gamma$ measures the maximum performance difference outside the majority group, and $\frac{\Gamma}{\rho}$ measures the "relative strength of minority variation to majority advantage." From Corollary 3.5, increasing $\pi_{\boldsymbol{\theta}}^\star$ effectively lowers the threshold of $\rho$ needed for homogeneous equilibrium. So even a small quality advantage $\rho$ on the dominant user type can outweigh potential gains from minority types (bounded by $\Gamma$), allowing a single model to dominate the entire market. To illustrate the parameter regime in which Corollary 3.5 applies, Fig. 3 plots the $\pi_{\boldsymbol{\theta}}^\star$ against $\frac{\Gamma}{\rho}$, the shaded region shows the parameter range where the homogeneous strategy $\boldsymbol{f}^* = (m, \ldots, m)$ is a PNE.

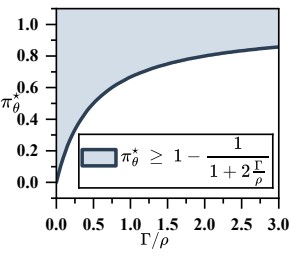

Figure 3: $\pi_{\boldsymbol{\theta}}^\star$ versus $\frac{\Gamma}{\rho}$.

## 4 USER WELFARE AND MARKET DIVERSITY

**Proposition 4.1.** *[Coverage Value Calculation] Given a strategy profile $\boldsymbol{f} = (f_1, \ldots, f_N)$, the coverage value in Definition 2.4 can be written as:*

$$V(\boldsymbol{f}) = \frac{1}{N} \sum_{i=1}^{N} \left( T_{f_i} + \delta_{f_i}(\boldsymbol{f}) \right)$$

*where $T$ and $\delta$ are defined in Definitions 3.1 and 3.2, respectively.*

**Scenario A:** fully differentiated equilibrium

$$\pi_{\boldsymbol{\theta}_A} = \pi_{\boldsymbol{\theta}_B} = 0.5$$

|  | $S_1(\boldsymbol{\theta})$ | $S_2(\boldsymbol{\theta})$ |
|---|---|---|
| $\boldsymbol{\theta}_A$ | 0.90 | 0.85 |
| $\boldsymbol{\theta}_B$ | 0.35 | 0.80 |

$$W = V(\boldsymbol{f}^*) = V(1,2) = 0.85$$

**Scenario B:** Add a new model $g_3$

$$\pi_{\boldsymbol{\theta}_A} = \pi_{\boldsymbol{\theta}_B} = 0.5$$

|  | $S_1(\boldsymbol{\theta})$ | $S_2(\boldsymbol{\theta})$ | $S_3(\boldsymbol{\theta})$ |
|---|---|---|---|
| $\boldsymbol{\theta}_A$ | 0.90 | 0.85 | 0.91 |
| $\boldsymbol{\theta}_B$ | 0.35 | 0.80 | 0.77 |

$$W = V(\boldsymbol{f}^*) = V(3,3) = 0.84$$

Figure 4: An example where enlarging the model pool decreases welfare. In Scenario A, with models $g_1$ and $d_2$ , the equilibrium is fully differentiated with the welfare $W = 0.85$. Adding a new model $g_3$ in Scenario B shifts the equilibrium to the homogeneous $(g_3, g_3)$, where welfare decreases to $W = 0.84$. It pulls both platforms toward homogenization, thereby sacrificing the welfare of minority types. The calculation details are provided in Section C.9.

Proposition 4.1 with proof in Section C.6 provides a closed-form expression for the coverage value of any strategy profile $\boldsymbol{f}$, showing that the sum of individual platform utilities equals the coverage value, i.e., $V(\boldsymbol{f}) = \sum_{i=1}^{N} U_i(\boldsymbol{f})$. However, under competition, self-interested platforms that each maximize their own utility $U_i$ do not necessarily achieve optimal user welfare, as discussed below.

**Lemma 4.2.** *Let $W$ denote the user welfare (Definition 2.5) achieved under the game $\mathcal{G}(\mathbb{G}, \mathbb{I}, \Theta)$, and $W_{\mathrm{opt}}$ the social optimum welfare (Definition 2.6). Then, it always holds that $W \leq W_{\mathrm{opt}}$.*

Note that the equality in Lemma 4.2 holds only in the degenerate cases: when $\boldsymbol{f}^* \in \arg\max_{\boldsymbol{f}} V(\boldsymbol{f})$ or when every $\boldsymbol{f}^{(t)}$ in the best-response cycle attains the maximum welfare value. Such situations rarely occur in competitive markets, highlighting the misalignment between platform incentives and user welfare as the social objective. We provide an example in Section C.8.

Next, we examine the impact of the number of models and platforms on the market. Intuitively, enlarging the model pool $\mathbb{G}$ or increasing the number of platforms $N$ might be expected to promote competition and enhance user welfare and market diversity. However, our counterexamples show that neither approach is reliably effective. As illustrated in Fig. 4, expanding the model pool can introduce a uniformly strong model, pulling the market toward homogenization and reducing welfare for minority users, as shown in Corollary 3.5. Similarly, adding platforms can be counterproductive: strategic interactions may induce best-response cycles or lead platforms to adopt weaker models to avoid competition, thereby lowering welfare. These results demonstrate that welfare and diversity are not monotone in competition intensity. Nonetheless, we can identify sufficient conditions under which platform entry does not reduce welfare or diversity, as detailed in Proposition 4.3.

**Proposition 4.3.** *Consider a game $\mathcal{G}(\mathbb{G}, \mathbb{I}, \Theta)$ with an equilibrium $\boldsymbol{f}^*$. Let $\widehat{\mathcal{G}}(\mathbb{G}, \mathbb{I}' := \mathbb{I} \cup \{i^+\}, \Theta)$ be another game with one additional platform added. Suppose there exists a model $h \in \mathbb{G}$ and an incumbent equilibrium strategy $\widehat{\boldsymbol{f}}$ from $\boldsymbol{f}^*$ such that the extended profile $\widehat{\boldsymbol{f}} := (\boldsymbol{f}^*, h)$ satisfies the best-response conditions: (i) the best response to $\boldsymbol{f}^*$ is $h$; (ii) no incumbent platform has a profitable deviation against $\widehat{\boldsymbol{f}}$. Then $\widehat{\boldsymbol{f}}$ is an equilibrium of the game $\widehat{\mathcal{G}}$. Furthermore, the user welfare and market diversity in $\widehat{\mathcal{G}}$ are at least as high as in $\mathcal{G}$, i.e., $\widehat{W} \geq W$ and $\widehat{D}_{\mathrm{supp}} \geq D_{\mathrm{supp}}$.*

## 5 FROM HARDMAX TO SOFTMAX USER CHOICE

Our main analysis adopts the hardmax user choice rule in Eq. 1, where each user deterministically selects the platform whose model achieves the highest score $S_{f_i}(\theta)$ for their type $\theta$, breaking ties uniformly. This assumption makes the analysis of strategic interactions more straightforward. In practice, however, users exhibit noisy and heterogeneous behavior rather than perfectly rational best responses. A natural extension is a softmax choice model. Given a profile $\boldsymbol{f} = (f_1, \ldots, f_N)$ and a user type $\boldsymbol{\theta}$, we define

$$p_i^{\mathrm{soft}}(\boldsymbol{\theta}) := \frac{e^{S_{f_i}(\boldsymbol{\theta})/\tau}}{\sum_{k=1}^{N} e^{S_{f_k}(\boldsymbol{\theta})/\tau}} \tag{8}$$

where $\tau \geq 0$ is a temperature parameter controlling the level of randomness in user choice. When $\tau \to \infty$, users are nearly indifferent and split across platforms almost uniformly. As $\tau$ decreases,

users concentrate more on higher-scoring platforms, and in the limit $\tau \to 0$ the softmax model converges to the hardmax rule.

Indeed, our negative result on the existence of equilibrium (Proposition 2.2) extends to the softmax user choice model. The platform game under Eq. 8 remains a finite normal-form game, and pure Nash equilibria may not exist. Proposition C.7 in Appendix C.10 shows that any instance with no PNE under the hardmax choice model remains without a PNE for all sufficiently small temperatures $\tau$ in the softmax model. We also provide an example in Appendix C.10 demonstrating that, for a fixed $\tau > 0$, the softmax model still admits no pure Nash equilibrium.

The utility decomposition (Proposition 3.3) and the existence of fully differentiated and homogeneous equilibrium (Lemma 3.4) also extend beyond the hardmax model. As shown in Appendix C.10, an analogous decomposition holds under softmax choice once the deviation term is redefined as $\delta_{f_i}^{\text{soft}}(\boldsymbol{f})$ in Eq. 21. With this modified deviation, Lemma 3.4 carries over directly: the equilibrium conditions retain the same form and can still be expressed as inequalities involving $T_{f_i}$ and $\delta_{f_i}^{\text{soft}}(\boldsymbol{f})$. This highlights that that market segmentation is not determined by average model performance alone; a model with lower average performance may support a differentiated equilibrium if it performs particularly well for certain user types.

Finally, our notion of welfare is largely independent of the choice rule. The only change is in the relation between coverage $V(\boldsymbol{f})$ and platform utilities: under hardmax, $\sum_i U_i(\boldsymbol{f}) = V(\boldsymbol{f})$, whereas under softmax $\sum_i U_i^{\text{soft}}(\boldsymbol{f}) \le V(\boldsymbol{f})$, typically with strict inequality, which further increases the misalignment between platform incentives and user welfare. Nevertheless, since the definition of coverage and welfare is determined only by the available models and their scores, all comparisons between $V(\boldsymbol{f})$ and $W_{\text{opt}}$, including Lemma 4.2 and its Proposition 4.3, remain valid.

## 6 DESIGNING COMPETITIVE MODELS FOR PLATFORM ADOPTION

In this section, we shift focus to model providers. Consider a single provider aiming to learn parameters $\phi$ for an entrant model $g_\phi$ that will be adopted by rational platforms. The provider seeks to maximize an adoption-weighted quality objective:

$$\max \ F(\phi) := \sum_{\boldsymbol{\theta} \in \Theta} \pi_{\boldsymbol{\theta}} \sigma_{\boldsymbol{\theta}} S_\phi(\boldsymbol{\theta})$$

where $\sigma_{\boldsymbol{\theta}} \in [0, 1]$ represents the adoption probability that users of type $\boldsymbol{\theta}$ would choose $g_\phi$ when it competes against incumbents, and $S_\phi(\boldsymbol{\theta}) = \mathbb{E}_{x \sim g_\phi}[r_{\boldsymbol{\theta}}(x)]$ is the expected quality that user type $\boldsymbol{\theta}$ receives from $g_\phi$.

To calculate $\sigma_{\boldsymbol{\theta}}$, suppose we can estimate the user distribution $\{\pi_{\boldsymbol{\theta}}\}_{\boldsymbol{\theta} \in \Theta}$ and the score $S_i(\boldsymbol{\theta})$ of incumbent $i \in \mathbb{M}$. Let $\bar{S}(\boldsymbol{\theta}) := \max_{j \in \mathbb{M}} S_j(\boldsymbol{\theta})$ be the best opponent score for user type $\boldsymbol{\theta}$. A hard adoption rule would set $\sigma_{\boldsymbol{\theta}} = 1$ when $S_\phi(\boldsymbol{\theta}) > \bar{S}(\boldsymbol{\theta})$ and 0 otherwise. As this is non-differentiable and unsuitable for gradient-based optimization, we adopt a Bradley-Terry Bradley & Terry (1952) soft gate on the margin $\Delta_{\boldsymbol{\theta}} := S_\phi(\boldsymbol{\theta}) - \bar{S}(\boldsymbol{\theta})$ and define the adoption probability of type $\boldsymbol{\theta}$ attracted by the new model as $\sigma_{\boldsymbol{\theta}} = \sigma(\beta \Delta_{\boldsymbol{\theta}})$, where $\sigma(z) = \frac{1}{1+e^{-z}}$. Here, $\beta$ controls the softness, as $\beta \to 0$, $\sigma_{\boldsymbol{\theta}}$ approaches the hard adoption.

We provide two solutions to solving the above optimization: 1) training data resampling; and 2) direct-gradient optimization.

**Training Data Resampling.** We first adopt a resampling-based scheme that biases the training data distribution toward user types with higher payoff weights $\alpha_{\boldsymbol{\theta}} := \pi_{\boldsymbol{\theta}} (\sigma_{\boldsymbol{\theta}})^\gamma \cdot \bar{S}(\boldsymbol{\theta})$, where $\gamma \ge 0$ emphasizes user types for which the entrant is more likely to outperform incumbents. Each data point $x$ is then assigned a sampling probability $\hat{w}(x)$, normalized from $w(x) \propto \sum_{\boldsymbol{\theta} \in \Theta} \alpha_{\boldsymbol{\theta}} v_{\boldsymbol{\theta}}(x)$, where $v_{\boldsymbol{\theta}}(x) \in [0, 1]$ measures how strongly $x$ is preferred by users of type $\boldsymbol{\theta}$. Specifically:

- **Structured data:** Each $x$ has an attribute (e.g., class, domain, style) $u(x) \in \mathbb{U}$, and type $\boldsymbol{\theta}$ specifies a distribution $q_{\boldsymbol{\theta}}(u)$. We set $v_{\boldsymbol{\theta}}(x) = q_{\boldsymbol{\theta}}(u(x))$, i.e., the probability that $x$ matches type $\boldsymbol{\theta}$'s preferred attribute. Sampling then proceeds by first drawing $u \sim \hat{w}(u) \propto \sum_{\boldsymbol{\theta}} \alpha_{\boldsymbol{\theta}} q_{\boldsymbol{\theta}}(u)$, and then sampling $x \sim D(\cdot \mid u)$.

- **Unstructured data:** We use the reward itself $v_{\boldsymbol{\theta}} = \text{normalize}(r_{\boldsymbol{\theta}}(x))$, so data points yielding higher expected rewards for type $\boldsymbol{\theta}$ are sampled more frequently.

In this method, type weights are computed based on $\sigma_{\boldsymbol{\theta}}$ and $\bar{S}(\boldsymbol{\theta})$, and the model $g_{\phi}$ is trained on data resampled according to these weights. This method alters the data distribution but not the loss function, making it compatible with standard training pipelines while effectively biasing training toward strategically valuable user types. The detailed procedure is provided in Algorithm 1.

**Direct-Gradient Optimization.** We train the model to directly improve both its generation quality and its competitive attractiveness against fixed opponents. Specifically, the training objective is:

$$\arg\min_{\phi} L(\phi) := \mathcal{L}(\phi) - \lambda F(\phi) \tag{9}$$

where $\mathcal{L}(\phi)$ is the standard loss ensuring that the model maintains overall sample quality, and $F(\phi)$ is the adoption-weighted quality objective that promotes competitiveness. The trade-off parameter $\lambda \geq 0$ balances quality and competitiveness. The main challenge in optimizing this objective via gradient descent lies in computing the gradient of $F(\phi)$. Note that the only term depending on $\phi$ in $F(\phi) = \sum_{\boldsymbol{\theta}\in\Theta} \pi_{\boldsymbol{\theta}}\sigma_{\boldsymbol{\theta}}S_{\phi}(\boldsymbol{\theta})$ is $S_{\phi}(\boldsymbol{\theta}) = \mathbb{E}_{x\sim g_{\phi}}[r_{\boldsymbol{\theta}}(x)]$. By the chain rule, we have:

$$\nabla_{\phi}F(\phi) = \sum_{\boldsymbol{\theta}\in\Theta} \pi_{\boldsymbol{\theta}}\left[\sigma_{\boldsymbol{\theta}} + \beta\sigma_{\boldsymbol{\theta}}\left(1 - \sigma_{\boldsymbol{\theta}}\right)S_{\phi}(\boldsymbol{\theta})\right] \cdot \nabla_{\phi}S_{\phi}(\boldsymbol{\theta})$$

We next present two estimators for $\nabla_{\phi}S_{\phi}(\boldsymbol{\theta})$. The detailed procedure is shown in Algorithm 2.

- **Pathwise gradient:** This estimator applies when both the reward function $r_{\boldsymbol{\theta}}(x)$ (e.g., classifier score, probability output) and the generative model (e.g., GAN (Goodfellow et al., 2014), DDPM (Ho et al., 2020), SGM (Song & Ermon, 2019)) are differentiable. The model samples $x = g_{\phi}(\xi)$, where $\xi \sim p_0(\xi)$ is drawn from a fixed prior $p_0$ and $g_{\phi}$ is a deterministic transformation of the noise $\xi$, then

$$\nabla_{\phi}S_{\phi}(\boldsymbol{\theta}) = \mathbb{E}_{\xi}\left[\nabla_{\mathrm{x}}r_{\boldsymbol{\theta}}(\mathrm{x})\Big|_{\mathrm{x}=g_{\phi}(\xi)} \cdot J_{\phi}g_{\phi}(\xi)\right]$$

where $J_{\phi}g_{\phi}(\xi)$ is the Jacobian of $g_{\phi}(\xi)$ with respect to $\phi$.

- **REINFORCE gradient** (Williams, 1992)**:** This estimator applies when either the reward function $r_{\boldsymbol{\theta}}(\mathrm{x})$ (e.g., discrete 0/1 feedback) or the generative process (e.g., SeqGAN (Yu et al., 2017), MaliGAN (Che et al., 2017))is non-differentiable. With a moving-average baseline $b_{\boldsymbol{\theta}}$ to reduce gradient variance and let $p_{\phi}(\mathrm{x})$ is the model distribution, then

$$\nabla_{\phi}S_{\phi}(\boldsymbol{\theta}) = \mathbb{E}_{\mathrm{x}\sim p_{\phi}}\left[(r_{\boldsymbol{\theta}}(\mathrm{x}) - b_{\boldsymbol{\theta}})\nabla_{\phi}\log(p_{\phi}(\mathrm{x}))\right]$$

# 7 EXPERIMENTS

In this section, we conduct experiments on both synthetic (Section D) and real-world data to provide a reproducible prototype for validating the theory [1].

**Model Pool.** We adopt a denoising diffusion probabilistic model (DDPM) (Ho et al., 2020) trained on the full CIFAR-10 dataset (Krizhevsky, 2009), contains 60,000 images from 10 classes $\mathbb{C} = \{\text{airplne} := 0, \text{automobile} := 1, \text{bird} := 2, \text{cat} := 3, \text{deer} := 4, \text{dog} := 5, \text{frog} := 6, \text{horse} := 7, \text{ship} := 8, \text{truck} := 9\}$, as the base model. To construct preference-oriented variants, we apply Low-Rank Adaptation (LoRA) (Hu et al., 2022) fine-tuning with different class-specific subsets. The choice of class groups and LoRA hyperparameters for each variant is summarized in Table 1. Each variant captures preferences aligned with a subset of CIFAR-10 classes.

**User Group.** We partition the user population into six groups, each characterized by heterogeneous preferences over CIFAR-10 classes. Formally, for user group $\boldsymbol{\theta}$, we specify a distribution of weights $\boldsymbol{\theta} = \{\theta_c\}_{c\in\mathbb{C}}$, $\sum_{c\in\mathbb{C}}\theta_c = 1$, the details are given in Table 2.

---

[1]Implementation available at: https://github.com/osu-srml/Generative_Competition

Table 1: Model Pool

| # | classes | $d$ | $\alpha_\ell$ | $\eta_\ell$ |
|---|---------|-----|----------|----------|
| M1 | airplane, auto | 4 | 16 | 1.0 |
| M2 | ship, truck | 4 | 16 | 1.0 |
| M3 | bird, cat | 4 | 16 | 1.0 |
| M4 | cat, dog | 8 | 32 | 1.5 |
| M5 | cat, dog | 4 | 16 | 1.0 |

Notes: $d$ denotes the LoRA rank, $\alpha_\ell$ is the LoRA scaling factor, and $\eta_\ell$ is the external scale applied during fine-tuning.

Table 2: User Groups

| #($\boldsymbol{\theta}$) | preferences(class($\theta_c$)) | $\pi(\boldsymbol{\theta})$ |
|---|---|---|
| A | cat (0.6), dog (0.4) | 0.18 |
| B | dog (0.7) cat (0.3) | 0.17 |
| C | airplane (0.5), ship (0.3), auto (0.2) | 0.16 |
| D | auto (0.6), truck (0.4) | 0.16 |
| E | bird (0.4), deer (0.3), frog (0.2), horse (0.1) | 0.17 |
| F | cat (0.2), dog (0.2), airplane (0.15), auto (0.15), ship (0.1), truck (0.2) | 0.16 |

**Reward Function.** We employ pretrained ResNet20 model (He et al., 2016) with the $92.60\%$ Top-1 accuary trained on CIFAR-10 and held fixed during experiments, assume $p_{\text{acc}}(c \mid \text{x})$ is the posterior class probability computed by this model for class $c$. Then the reward of v for user type $\boldsymbol{\theta}$ is calculated by $r_{\boldsymbol{\theta}}(x) = \sum_{c\in\mathbb{C}} \theta_c \cdot p_{\text{acc}}(c \mid \text{x})$. For every calculation of $S$, we sample 2000 samples.

**Discrete Best-Response Simulation.** The average performance of the five models $T_i$ and their user-specific performance $S_i(\boldsymbol{\theta})$ are shown in Fig. 8 in Section E.1, where we conduct a discrete best-response simulation by progressively enlarging the model pool (from 1 to 5 models with 3 players) and increasing the number of platforms (from 1 to 6 players with 5 models). At each round, platforms update their strategies by choosing the best response among the available models, given the current distribution of opponents' choices. For each game, we perform three independent runs and track diversity $D_{\text{HHI}}$ and coverage value $V(\boldsymbol{f})$ at every step $t$ of best-response dynamics.

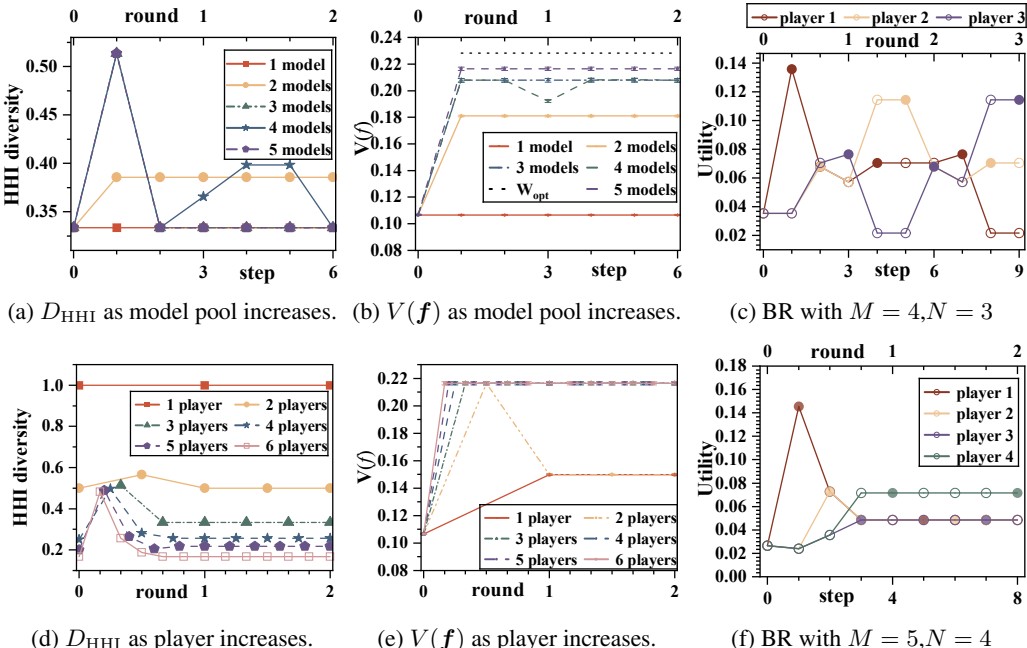

(a) $D_{\text{HHI}}$ as model pool increases. (b) $V(\boldsymbol{f})$ as model pool increases. (c) BR with $M=4, N=3$

(d) $D_{\text{HHI}}$ as player increases. (e) $V(\boldsymbol{f})$ as player increases. (f) BR with $M=5, N=4$

Figure 5: When enlarging either the model pool (a,b) with 3 players or the number of platforms (d,e) with 5 models, the change of HHI diversity ((a,d), where larger values indicate more homogenization) and coverage value ((b,e), where larger values are better). (c,f) provide examples of utility trajectories across best-response steps, where filled markers denote the player taking the action at each step. (c) shows best-response cycle and (d) shows an equilibrium.

From the Fig. 5a, enlarging the model pool does not automatically increase diversity. Only when the newly introduced models are sufficiently distinct (add M2 or M3) can they enhance diversity. By contrast, strong entrants that are merely close substitutes for existing leaders tend to homogenize the market (add M5), as platforms converge toward the same high-performing options. This reproduces

the convergence phenomenon widely observed in today's generative model markets. Increasing the number of platforms in Fig. 5d, however, expands adoption opportunities and thus promotes diversity. In terms of welfare, adding more platforms as Fig. 5e (which enables greater choice) improves user welfare and accelerates its growth. However, welfare never reaches the social optimum. The trajectory examples in Fig. 5c and Fig. 5f further reveal that early movers often select the "best" model, but are later forced to share its benefits with subsequent players, leaving their utilities suboptimal. In contrast, players who move later sometimes adopt models that are less attractive globally but provide relative advantages when not shared, resulting in higher individual utilities.

In Section D, we systematically vary the model pool size, the number of platform players, and the user group distributions on synthetic data, with detailed results and figures provided.

**Algorithmic Best-Response Entry.** The hyperparameters and full algorithmic details are provided in the Section. E.2.1. Section E.2.2 reports a systematic hyperparameter tuning for both methods. Both algorithms are initialized on the full CIFAR-10 dataset. The result is shown in Fig. 6.

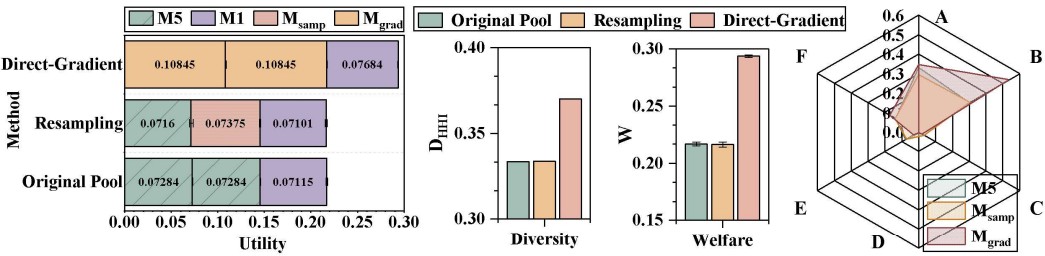

Figure 6: Performance of new models generated by the direct-gradient method $M_{\text{grad}}$ and the resampling method $M_{\text{samp}}$. (a) Left: equilibrium outcomes and utilities of three players when competing over the original pool, after introducing $M_{\text{grad}}$ and $M_{\text{samp}}$. (b) Middle: diversity $D_{\text{HHI}}$ and welfare $W_{\text{eq}}$ under the three equilibrium. (c) Right: comparison between the best original model $M_5$ and the new entrants $M_{\text{grad}}$, $M_{\text{samp}}$ on user scores.

The direct-gradient method achieves stronger performance: it successfully replaces the best model in the original pool, dominates the market, and yields higher welfare and diversity. Moreover, it converges with fewer iterations ($\approx 20$). However, it requires modifying the model's internal objective, and the diversity of sample classes is decreased as shown in Fig. 9.

The resampling method suffers from higher variance due to stochasticity. It only approaches the best model in the original pool, while reducing welfare. It is also more computationally demanding ($\approx$ 10 resampling with 50 iterations each). However, the method has the practical advantage of being plug-and-play: it can be applied to any model without altering its loss function.

## 8 CONCLUSION

In this paper, we formalize generative AI markets as a three-layer model-platform-user game. From the platform perspective, we characterize conditions for both fully differentiated and homogeneous equilibria, showing that the market is jointly shaped by average model performance and user-specific deviation advantages. From the user perspective, we demonstrate that enlarging the model pool or increasing the number of platforms does not necessarily translate into higher welfare. From the model provider perspective, we propose training schemes that strategically facilitate entry into competitive markets. Together, these findings highlight inherent paradoxes in generative AI markets and point to design principles for more socially aligned ecosystems.

## ETHICS STATEMENT

This work analyzes competitive generative model markets using both theoretical modeling and empirical experiments. Our analysis is abstracted from specific settings and does not involve sensitive

personal data, human subjects, or system manipulations. Our findings raise broader ethical implications. The results show that competition among generative models may reduce diversity and user welfare, highlighting the need for responsible governance and transparent platform practices. The proposed training schemes are designed to advance understanding of market dynamics, not to prescribe adversarial practices. Overall, this work aims to inform policy discussions and support the design of more socially beneficial generative ecosystems.

## REPRODUCIBILITY STATEMENT

We ensure reproducibility by including all definitions, propositions, and proofs, and by detailing model pool construction, user groups, reward functions, and evaluation metrics. Hyperparameters and training procedures are documented in the appendix, and code is released to support replication and extension of our results.

## ACKNOWLEDGMENTS

This work was funded in part by the National Science Foundation under award number IIS-2202699, IIS-2416895, IIS-2301599, and CMMI-2301601.

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

# A  RELATED WORK

Recent advances in machine learning has led to the emergence of markets in which multiple models simultaneously operate and compete for user adoption. In such competitive environments, the strategic interactions among providers critically shape both market outcomes and overall social welfare.

**Competition in Predictive Models.** Competition among classification models has been extensively studied through the market design and strategic learning. A key insight is that in competitive markets, maximizing classification accuracy alone does not guarantee higher adoption or improved social welfare. For example, recent work Einav & Rosenfeld (2025) formalizes the accuracy market, where multiple classification providers compete for users, showing that optimal strategies must account for rivals' actions rather than accuracy in isolation. Similarly, Ben-Porat & Tennenholtz (2017) study the learnability of optimal responses in competitive regression, and further establish that pure-strategy equilibria exist and that competition may induce strategic misprediction (Ben-Porat & Tennenholtz, 2019). Further, Jagadeesan et al. (2023b) demonstrate that even when individual predictors achieve lower Bayes risk, strategic competition can paradoxically reduce overall social welfare. Extending beyond classification and regression, Yao et al. (2023) analyze how top-K recommendation performs under competing content creators, showing that user welfare losses remain bounded, while Yao et al. (2024b) propose platform interventions that directly optimize user welfare in such competitive recommendation environments. Overall, these results show that accuracy must be assessed in the context of competition, entry, and social welfare.

**Competition in Generative Models.** As model capabilities improve, research has increasingly focused on competition among generative models. Raghavan (2024) show that equilibrium under generative AI competition tends toward content homogeneity, even when models perform well in isolation, while stronger competition can counteract this effect. Empirical studies further suggest that generative AI usage in areas such as peer review (Ebadi et al., 2025; Kankanhalli, 2024), writing (Doshi & Hauser, 2024), and creative generation (Wu et al., 2025) often associated with reduced output diversity. Beyond direct competition between models, recent work also investigates interactions between generative AI and human participants. At the creator level, Yao et al. (2024a) model competition between human creators and generative AI using a generalized contest framework, showing conditions for coexistence, conflict, or even the absence of stable equilibria; while at the platform level, Taitler & Ben-Porat (2025) demonstrate that generative AI can paradoxically reduce overall welfare in human-driven platforms, echoing Braess's paradox.

**Strategic Model Behavior.** Another line of work studies how learning systems are shaped by strategic environments and feedback-driven dynamics. For example, Wei & Zhang (2025) show that when models learn from feedback, they can infer latent human preferences from the resulting data and manipulate the feedback signals, inducing undesirable shifts in downstream output distributions. Related concerns also arise in performative prediction (Perdomo et al., 2020; Jin et al., 2026) and strategic learning (Zuo et al., 2024; Xie & Zhang, 2024b;a; Jin et al., 2022; Zhang et al., 2022), where model deployment endogenously alters the data-generating process, inducing distributional drift and welfare loss.

Unlike prior studies that focus on two-layer market, our work formalizes a three-layer model-platform-user game. Under the assumption of deterministic user choice, we show that pure Nash equilibria may not exist. Building on this observation, we characterize the conditions under which equilibria arise and analyze how the resulting market structures shape welfare and diversity as the pool of available models expands. Moreover, we depart from prior work by adopting the perspective of model providers and proposing best-response entry training schemes that allow entrants to strategically introduce new models. This provider-centric view and the resulting entry dynamics are largely absent from existing studies of model competition.

# B  DISCUSSION

**Platforms with Multiple Models.** For tractability, the current framework assumes that each platform selects a single model. However, it can be extended naturally to accommodate multiple models per platform. In this setting, each platform's strategy would be a set of models $M_i$, and user choice could depend on the highest-performing model in that set for their type, $\hat{S}_i(\theta) = \max_{j \in M_i} S_j(\theta)$,

or an expected score $\hat{S}_i(\theta) = \mathbb{E}_{j \in M_i} S_j(\theta)$. The three-layer market formalization remains as before, with platform payoffs computed using $\hat{S}_i(\theta)$ instead of $S_i(\theta)$, and best responses now taken over model mixtures rather than single models. Then, many of the existing analysis, such as the utility decomposition, equilibrium characterization, and welfare analysis, can be generalized to this setting.

**Partially Overlapping Markets.** Our framework is designed for settings where platforms offer comparable services and draw from a shared pool of models $M$. In such markets, platforms face similar types of demand (e.g., overlapping mixes of coding and translation tasks), and their strategic decision is which model from this common pool to deploy in order to attract groups. By contrast, if different platforms specialize in largely disjoint services (e.g., one focuses almost exclusively on coding while another focuses almost exclusively on translation), then their effective model pools may overlap only partially or not at all. In that case, they are not competing for the same user base in the sense of our model: a user might naturally use both services for different tasks, and platforms no longer face a shared competitive environment. Extending the framework to this multi-market or partially overlapping-market environments is an interesting direction for future work.

**Dynamic Interactions.** Our analysis focuses on first-round interactions in a static, complete-information setting. In practice, user behavior and platform strategies evolve gradually: the data distribution shifts under repeated deployments, and models are retrained iteratively. Incorporating these multi-round feedback effects into our three-layer framework is left for future work.

## C    PROOFS

### C.1    PROOF OF PROPOSITION 2.2

**Proposition 2.2.** *[Nonexistence of PNE] Consider the game $\mathcal{G}(\mathbb{G}, \mathbb{I}, \Theta)$ with finite sets of platforms $\mathbb{I}$, models $\mathbb{G}$, and user types $\Theta$, where each platform $i$ chooses a model $f_i \in \mathbb{M}$ based on Eq. 1. The game may not admit a pure-strategy Nash equilibrium $\boldsymbol{f}^*$.*

*Proof.* We provide a constructive counterexample here. Let $\Theta = \{\boldsymbol{\theta}_A, \boldsymbol{\theta}_B, \boldsymbol{\theta}_c\}$ with uniform weights $\pi(\boldsymbol{\theta}_k) = \frac{1}{3}$. Let $\mathbb{G} = \{g_1, g_2, g_3\}$ and define scores:

|  | $S_1(\boldsymbol{\theta})$ | $S_2(\boldsymbol{\theta})$ | $S_3(\boldsymbol{\theta})$ |
|---|---|---|---|
| $\boldsymbol{\theta}_A$ | 0.2 | 0.1 | 0 |
| $\boldsymbol{\theta}_B$ | 0 | 0.2 | 0.1 |
| $\boldsymbol{\theta}_C$ | 0.1 | 0 | 0.2 |

Then when there are two players, the payoff matrix is :

| $\boldsymbol{f} = (f_1, f_2)$ | $g_1$ | $g_2$ | $g_3$ |
|---|---|---|---|
| $g_1$ | $(0.05, 0.05)$ | $(0.1, 0.067)$ | $(0.067, 0.1)$ |
| $g_2$ | $(0.067, 0.1)$ | $(0.05, 0.05)$ | $(0.1, 0.067)$ |
| $g_3$ | $(0.1, 0.067)$ | $(0.067, 0.1)$ | $(0.05, 0.05)$ |

On the diagonal $(g_k, g_k)$ each platform gets $0.05$. Against $g_k$, the unique best response is the model that yields $0.10$, so any diagonal profile is profitably deviated from. Off the diagonal, the player receiving $0.067$ can switch to the third model and improve to $0.10$. Hence no profile is a mutual best response; therefore no PNE exists. □

### C.2    PROOFS OF PROPOSITION 3.3

To illustrate the intuition, we first consider the case with two platforms $N = 2$.

The strategy profile is $\boldsymbol{f} = (f_i, f_j)$, in this case, the attraction term in Definition 3.2 simplifies to:

$$Z_{ij}(\boldsymbol{\theta}) := \begin{cases} S_{f_i}(\boldsymbol{\theta}) & \text{if } S_{f_i}(\boldsymbol{\theta}) > S_{f_j}(\boldsymbol{\theta}) \\ 0 & \text{if tie} \\ -S_{f_i}(\boldsymbol{\theta}) & \text{if } S_{f_i}(\boldsymbol{\theta}) < S_{f_j}(\boldsymbol{\theta}) \end{cases} \tag{10}$$

which measures how much user type $\boldsymbol{\theta}$ strictly prefers $f_i$ over $f_j$. Accordingly, the deviation advantage is:

$$\delta_{ij} := \sum_{\boldsymbol{\theta} \in \Theta} \pi(\boldsymbol{\theta}) \cdot Z_{ij}(\boldsymbol{\theta}) \tag{11}$$

**Proposition C.1** (Utility Decomposition for two platforms.)**.** *Suppose* $N = 2$, *let* $i, j \in \mathbb{I}$ *be the two players who choose model* $f_i$ *and* $f_j$ *under strategy* $\boldsymbol{f}$. *Then the expected utility is*

$$U_i(f_i, f_j) = \begin{cases} \frac{1}{2} T_{f_i} & \text{if } f_i = f_j \\ \frac{1}{2}(T_{f_i} + \delta_{ij}) & \text{if } f_i \neq f_j \end{cases} \tag{12}$$

*Proof.* Consider the case with two platforms ($N = 2$) and strategy profile $\boldsymbol{f} = (f_i, f_j)$, let $Win_i = \{\boldsymbol{\theta} : S_{f_i}(\boldsymbol{\theta}) > S_{f_j}(\boldsymbol{\theta})\}$, $Tie = \{\boldsymbol{\theta} : S_{f_i}(\boldsymbol{\theta}) = S_{f_j}(\boldsymbol{\theta})\}$, $Loser_i = \{\boldsymbol{\theta} : S_{f_i}(\boldsymbol{\theta}) < S_{f_j}(\boldsymbol{\theta})\}$.

With Eq. 1, a type $\boldsymbol{\theta}$ is fully assigned to the winner, split evenly on a tie, and assigned zero to the loser. Therefore, the utility of the platform choosing $g_i$ is:

$$U_i(f_i, \boldsymbol{f}_{-i}) = \sum_{\boldsymbol{\theta} \in \Theta} \pi_{\boldsymbol{\theta}} \cdot p_i(\boldsymbol{\theta}) \cdot S_{f_i}(\boldsymbol{\theta}) = \sum_{\boldsymbol{\theta} \in Win_i} \pi(\boldsymbol{\theta}) S_{f_i}(\boldsymbol{\theta}) + \frac{1}{2} \sum_{\boldsymbol{\theta} \in Tie} \pi(\boldsymbol{\theta}) S_{f_i}(\boldsymbol{\theta})$$

By definition,

$$T_i = \sum_{\boldsymbol{\theta} \in \Theta} \pi(\boldsymbol{\theta}) S_{f_i}(\boldsymbol{\theta}) = \sum_{\boldsymbol{\theta} \in Win_i} \pi(\boldsymbol{\theta}) S_{f_i}(\boldsymbol{\theta}) + \sum_{\boldsymbol{\theta} \in Tie} \pi(\boldsymbol{\theta}) S_{f_i}(\boldsymbol{\theta}) + \sum_{\boldsymbol{\theta} \in Loser_i} \pi(\boldsymbol{\theta}) S_{f_i}(\boldsymbol{\theta})$$

$$\delta_{ij} = \sum_{\boldsymbol{\theta} \in \Theta} \pi(\boldsymbol{\theta}) Z_{ij}(\boldsymbol{\theta}) = \sum_{\boldsymbol{\theta} \in Win_i} \pi(\boldsymbol{\theta}) S_{f_i}(\boldsymbol{\theta}) - \sum_{\boldsymbol{\theta} \in Loser_i} \pi(\boldsymbol{\theta}) S_{f_i}(\boldsymbol{\theta})$$

Adding these two:

$$T_{f_i} + \delta_{ij} = 2 \sum_{\boldsymbol{\theta} \in Win_i} \pi(\boldsymbol{\theta}) S_{f_i}(\boldsymbol{\theta}) + \sum_{\boldsymbol{\theta} \in Tie} \pi(\boldsymbol{\theta}) S_{f_i}(\boldsymbol{\theta})$$

Then

$$\frac{1}{2}(T_{f_i} + \delta_{ij}) = \sum_{Win_i} \pi(\boldsymbol{\theta}) S_{f_i}(\boldsymbol{\theta}) + \frac{1}{2} \sum_{T} \pi(\boldsymbol{\theta}) S_{f_i}(\boldsymbol{\theta}) = u_i$$

If both platforms choose the same model, then all users are split evenly, so

$$u_i = u_j = \frac{1}{2} \sum_{\boldsymbol{\theta}} \pi(\boldsymbol{\theta}) S_{f_i}(\boldsymbol{\theta}) = \frac{1}{2} T_{f_i}.$$

Therefore, for $N = 2$:

$$\boxed{U_i(f_i, f_j) = \begin{cases} \frac{1}{2} T_{f_i} & \text{if } f_i = f_j \\ \frac{1}{2}(T_{f_i} + \delta_{ij}) & \text{if } f_i \neq f_j \end{cases}}$$

$\square$

**Proposition 3.3.** *[Utility Decomposition.]* *The expected utility* $U_i(f_i; \boldsymbol{f}_{-i})$ *of platform* $i$ *in Eq. 2 can be decomposed into* $T_{f_i}$ *and* $\delta_{f_i}(\boldsymbol{f})$ *as:*

$$U_i(f_i; \boldsymbol{f}_{-i}) = \frac{1}{N}(T_{f_i} + \delta_{f_i}(\boldsymbol{f})). \tag{5}$$

*Proof.* Fix a strategy profile $\boldsymbol{f} = (f_1, \ldots, f_N)$ and a model $f_i$ chosen by player $i$. Recall that let $\mathbb{M}(\boldsymbol{f}) = \{f_1, \ldots, f_N\}$ denote the set of models used in this strategy. For a user type $\boldsymbol{\theta}$, define the set of maximizers $\mathbb{A}_{\boldsymbol{f}}(\boldsymbol{\theta}) := \arg\max_{k \in \mathbb{M}(\boldsymbol{f})} S_k(\boldsymbol{\theta})$ and let $A_{\boldsymbol{f}}(\boldsymbol{\theta}) := |\mathbb{A}_{\boldsymbol{f}}(\boldsymbol{\theta})|$ be the number of models tied for the maximum.

Under the rule, the share of type $\boldsymbol{\theta}$ that is allocated to a platform using $f_i$ is

$$p_i(\boldsymbol{\theta}; \boldsymbol{f}) := \begin{cases} \frac{1}{A_{\boldsymbol{f}}(\boldsymbol{\theta})} & f_i \in \mathbb{A}_{\boldsymbol{f}}(\boldsymbol{\theta}) \\ 0 & f_i \notin \mathbb{A}_{\boldsymbol{f}}(\boldsymbol{\theta}). \end{cases}$$

Hence the expected utility of player $i$ equals

$$U_i(f_i, \boldsymbol{f}_{-i}) = \sum_{\boldsymbol{\theta} \in \Theta} \pi(\boldsymbol{\theta}) p_i(\boldsymbol{\theta}; \boldsymbol{f}) S_{f_i}(\boldsymbol{\theta}).$$

We now prove the following per-type identity:

$$N \cdot p_j(\boldsymbol{\theta}; \boldsymbol{f}) \cdot S_{f_j}(\boldsymbol{\theta}) = S_{f_j}(\boldsymbol{\theta}) + Z_j(\boldsymbol{\theta}; \boldsymbol{f}) \quad \forall \boldsymbol{\theta} \in \Theta \tag{13}$$

where $Z_j(\boldsymbol{\theta}; \boldsymbol{f})$ is defined in Definition 3.2.

**Case 1:** $f_j \notin \mathbb{A}_{\boldsymbol{f}}(\boldsymbol{\theta})$. Then $p_j(\boldsymbol{\theta}; \boldsymbol{f}) = 0$, so the left-hand side of Eq. 13 is 0. By Definition 3.2, $Z_j(\boldsymbol{\theta}; \boldsymbol{f}) = -S_{f_j}(\boldsymbol{\theta})$, hence $S_{f_j}(\boldsymbol{\theta}) + Z_j(\boldsymbol{\theta}; \boldsymbol{f}) = S_{f_j}(\boldsymbol{\theta}) - S_{f_j}(\boldsymbol{\theta}) = 0$. Thus Eq. 13 holds.

**Case 2:** $j \in \mathbb{A}_{\boldsymbol{f}}(\boldsymbol{\theta}))$. Then $p_j(\boldsymbol{\theta}; \boldsymbol{f}) = \frac{1}{A_{\boldsymbol{f}}(\boldsymbol{\theta})}$. Again by Definition 3.2

$$Z_j(\boldsymbol{\theta}; \boldsymbol{f}) = \frac{N - A_{\boldsymbol{f}}(\boldsymbol{\theta})}{A_{\boldsymbol{f}}(\boldsymbol{\theta})} S_{f_j}(\boldsymbol{\theta})$$

Therefore

$$S_{f_j}(\boldsymbol{\theta}) + Z_j(\boldsymbol{\theta}; \boldsymbol{f}) = \left(1 + \frac{N - A_{\boldsymbol{f}}(\boldsymbol{\theta})}{A_{\boldsymbol{f}}(\boldsymbol{\theta})}\right) S_{f_j}(\boldsymbol{\theta}) = \frac{N}{A_{\boldsymbol{f}}(\boldsymbol{\theta})} S_{f_j}(\boldsymbol{\theta}) = N p_j(\boldsymbol{\theta}; \boldsymbol{f}) S_{f_j}(\boldsymbol{\theta})$$

So Eq. 13 also holds.

When we have Eq. 13, sum both sides over $\boldsymbol{\theta}$ with weights $\pi(\boldsymbol{\theta})$ and divide by $N$:

$$\sum_{\boldsymbol{\theta} \in \Theta} \pi(\boldsymbol{\theta}) p_j(\boldsymbol{\theta}; \boldsymbol{f}) S_{f_j}(\boldsymbol{\theta}) = \frac{1}{N} \sum_{\boldsymbol{\theta} \in \Theta} \pi(\boldsymbol{\theta}) \left(S_{f_j}(\boldsymbol{\theta}) + Z_j(\boldsymbol{\theta}; \boldsymbol{f})\right) = \frac{1}{N} \left(T_{f_j} + \delta_{f_j}(\boldsymbol{f})\right)$$

where $\delta_j(\boldsymbol{f}) := \sum_{\boldsymbol{\theta}} \pi(\boldsymbol{\theta}) Z_j(\boldsymbol{\theta}; \boldsymbol{f})$.

Since $U_i(f_i, \boldsymbol{f}_{-i}) = \sum_{\boldsymbol{\theta} \in \Theta} \pi(\boldsymbol{\theta}) p_i(\boldsymbol{\theta}; \boldsymbol{f}) S_{f_i}(\boldsymbol{\theta})$, we obtain:

$$\boxed{U_i(f_i, \boldsymbol{f}_{-i}) = \frac{1}{N} \left(T_{f_i} + \delta_{f_i}(\boldsymbol{f})\right)}$$

$\square$

## C.3 Proofs of Lemma 3.4

We first consider the case with $N = 2$.

**Lemma C.2** (Conditions for Equilibrium for two platforms). *Consider a game with $N = 2$ platform players choosing between $M$ models $\mathbb{G}$ with a finite users' type space $\Theta$ with weights $\pi(\boldsymbol{\theta}) \geq 0$, $\sum_{\boldsymbol{\theta} \in \Theta} \pi(\boldsymbol{\theta}) = 1$. The utility of each player is defined in Eq. 2. A strategy $\boldsymbol{f}^* = (f_1^* = g_i, f_2^* = g_j)$ with $i \neq j$ is a **fully differentiated equilibrium** iff*

$$\begin{cases} T_i + \delta_{ij} \geq \max\{T_j, \max_{k \neq j}\{T_k + \delta_{kj}\}\} \\ T_j + \delta_{ji} \geq \max\{T_i, \max_{k \neq i}\{T_k + \delta_{ki}\}\} \end{cases} \tag{14}$$

*When $M = 2$, the condition becomes*

$$-\delta_{ij} \leq T_i - T_j \leq \delta_{ji} \tag{15}$$

*A strategy $\boldsymbol{f}^* = (f_1^*, f_2^*)$ is a **homogeneous equilibrium** where all $f_i^* = m$ for some $m \in \mathbb{M}$ iff*

$$\exists m \in M \text{ s.t. } T_m - T_k \geq \delta_{km} \quad \forall k \in \mathbb{M} \setminus \{m\} \tag{16}$$

*When $M = 2$, the condition becomes*

$$T_j - T_i > \delta_{ij} \quad \text{or} \quad T_i - T_j > \delta_{ji} \tag{17}$$

*Proof.* First, let's consider that there is only two models, $N = 2$ and $M = 2$. Using Proposition C.1, we obtain the utility of each model, from which the payoff matrix can be derived.

| $\boldsymbol{f}$ | $g_i$ | $g_j$ |
|---|---|---|
| $g_i$ | $\left(\frac{1}{2}T_i, \frac{1}{2}T_i\right)$ | $\left(\frac{1}{2}(T_i + \delta_{ij}), \frac{1}{2}(T_j + \delta_{ji})\right)$ |
| $g_j$ | $\left(\frac{1}{2}(T_j + \delta_{ji}), \frac{1}{2}(T_i + \delta_{ij})\right)$ | $\left(\frac{1}{2}T_j, \frac{1}{2}T_j\right)$ |

Suppose players choose different models. This is a pure strategy Nash equilibrium if and only if neither player wants to deviate, that is:

$$\begin{cases} \frac{1}{2}(T_i + \delta_{ij}) \geq \frac{1}{2}T_j \\ \frac{1}{2}(T_j + \delta_{ji}) \geq \frac{1}{2}T_i \end{cases} \quad \Longleftrightarrow \quad \begin{cases} T_i + \delta_{ij} \geq T_j \\ T_j + \delta_{ji} \geq T_i \end{cases}$$

Then the condition is:

$$\boxed{-\delta_{ij} \leq T_i - T_j \leq \delta_{ij}}$$

Suppose players choose the same model. This is a pure strategy Nash equilibrium if and only if neither player wants to deviate:

$$\frac{1}{2}(T_j + \delta_{ji}) < \frac{1}{2}T_i \quad \Longleftrightarrow \quad T_j + \delta_{ji} < T_i$$

or

$$\frac{1}{2}(T_i + \delta_{ij}) < \frac{1}{2}T_j \quad \Longleftrightarrow \quad T_i + \delta_{ij} < T_j$$

Then the condition is:

$$\boxed{T_j - T_i > \delta_{ij} \quad \text{or} \quad T_i - T_j > \delta_{ji}}$$

Second, let's consider that there is more than two models. The payoff matrix is:

| $\boldsymbol{f}$ | $g_i$ | $g_j$ | $\cdots$ | $g_k$ |
|---|---|---|---|---|
| $g_i$ | $\left(\frac{1}{2}T_i, \frac{1}{2}T_i\right)$ | $\left(\frac{1}{2}(T_i + \delta_{ij}), \frac{1}{2}(T_j + \delta_{ji})\right)$ | $\cdots$ | $\left(\frac{1}{2}(T_i + \delta_{ik}), \frac{1}{2}(T_k + \delta_{ki})\right)$ |
| $g_j$ | $\left(\frac{1}{2}(T_j + \delta_{ji}), \frac{1}{2}(T_i + \delta_{ij})\right)$ | $\left(\frac{1}{2}T_j, \frac{1}{2}T_j\right)$ | $\cdots$ | $\left(\frac{1}{2}(T_j + \delta_{jk}), \frac{1}{2}(T_k + \delta_{kj})\right)$ |
| $\cdots$ | $\cdots$ | $\cdots$ | $\cdots$ | $\cdots$ |
| $g_k$ | $\left(\frac{1}{2}(T_k + \delta_{ki}), \frac{1}{2}(T_i + \delta_{ij})\right)$ | $\left(\frac{1}{2}(T_k + \delta_{kj}), \frac{1}{2}(T_j + \delta_{jk})\right)$ | $\cdots$ | $\left(\frac{1}{2}T_k, \frac{1}{2}T_k\right)$ |

Suppose players choose different models $i, j$. This is a pure strategy Nash equilibrium if and only if neither player wants to deviate, that is:

$$\begin{cases} \frac{1}{2}(T_i + \delta_{ij}) \geq \frac{1}{2}T_j \\ \frac{1}{2}(T_i + \delta_{ij}) \geq \frac{1}{2}(T_k + \delta_{kj}) \quad \forall k \in M \\ \frac{1}{2}(T_j + \delta_{ji}) \geq \frac{1}{2}T_i \\ \frac{1}{2}(T_j + \delta_{ji}) \geq \frac{1}{2}(T_k + \delta_{ik}) \quad \forall k \in M \end{cases} \quad \Longleftrightarrow \quad \begin{cases} T_i + \delta_{ij} \geq T_j \\ T_j + \delta_{ji} \geq T_i \end{cases}$$

Then the condition is:

$$\boxed{\exists i \neq j \in M \text{ s.t.} \begin{cases} T_i + \delta_{ij} \geq \max\{T_j, \max_{k \neq j}\{T_k + \delta_{kj}\}\} \\ T_j + \delta_{ji} \geq \max\{T_i, \max_{k \neq i}\{T_k + \delta_{ki}\}\} \end{cases}}$$

Suppose players choose different models $m$. This is a pure strategy Nash equilibrium if and only if neither player wants to deviate:

$$\frac{1}{2}(T_j + \delta_{ji}) < \frac{1}{2}T_i \quad \Longleftrightarrow \quad T_j + \delta_{ji} < T_i \quad \forall i \neq j$$

Then the condition is:

$$\boxed{\exists m \in M \text{ s.t.} \quad T_m - T_k \geq \delta_{km} \quad \forall k \in M \setminus \{m\}}$$

$\square$

**Lemma 3.4.** *[Existence of Equilibrium] Consider the game $\mathcal{G}(\mathbb{G}, \mathbb{I}, \Theta)$ with finite user types $\Theta$, and $N$ platforms choosing from $M$ models $\mathbb{G}$, where $M \geq N$. A **fully differentiated equilibrium** $\boldsymbol{f}^* = (f_1^*, \ldots, f_N^*)$ exists if and only if for every platform $i$ and every alternative model $f_i \in \mathbb{G} \setminus \{f_i^*\}$,*

$$T_{f_i^*} - T_{f_i} \geq \delta_{f_i}(\boldsymbol{f}_{-i}^* \cup f_i) - \delta_{f_i^*}(\boldsymbol{f}^*) \tag{6}$$

*A **homogeneous equilibrium** $\boldsymbol{f}^* = (f_1^*, \ldots, f_N^*)$, $f_i^* = m$ exists if and only if for some $m \in \mathbb{M}$,*

$$T_m - T_{f_i} \geq \delta_{f_i}(\boldsymbol{f}_{-m}^* \cup f_i) - \delta_m(\boldsymbol{f}^*) \tag{7}$$

*Proof.* Fix a candidate profile $\boldsymbol{f}^* = (f_1^*, \ldots, f_N^*)$. For any player $i$ and any deviation $g \in \mathbb{G} \setminus \{f_i^*\}$, let $\boldsymbol{f}' = (f_1^*, \ldots, f_{i-1}^*, g, f_{i+1}^*, \ldots, f_N^*) = \boldsymbol{f}_{-i}^* \cup \{g\}$.

By definition of a pure Nash equilibrium, $\boldsymbol{f}^*$ is a PNE iff for all $i$ and all such $g$,

$$U_i(\boldsymbol{f}^*) \geq U_i(\boldsymbol{f}')$$

Using Proposition 3.3, we have

$$N \cdot U_i(\boldsymbol{f}^*) = T_{f_i^*} + \delta_{f_i^*}(\boldsymbol{f}^*) \qquad N \cdot U_i(\boldsymbol{f}') = T_g + \delta_g(\boldsymbol{f}')$$

Therefore:

$$T_{f_i^*} + \delta_{f_i^*}(\boldsymbol{f}^*) \geq T_g + \delta_g(\boldsymbol{f}')$$

Conversely, if it holds for all $i$ and all $g \neq f_i^*$, then the above inequality reverses to $U_i(\boldsymbol{f}^*) \geq U_i(\boldsymbol{f}')$ for every deviation, so no player profits from deviating and $\boldsymbol{f}^*$ is a PNE. This completes the proof of the fully differentiated case.

The homogeneous case is similar, with $f_i^* = m$ for all $i$; plugging $m$ into the same inequality we obtain the desired results. □

### C.4 CALCULATION OF THE EXAMPLE IN FIG. 2

**Scenario A:** Two user types $\Theta = \{\boldsymbol{\theta}_A, \boldsymbol{\theta}_B\}$ with equal weights $\pi(\boldsymbol{\theta}_A) = \pi(\boldsymbol{\theta}_B) = 0.5$. The model scores are:

| | $S_1(\boldsymbol{\theta})$ | $S_2(\boldsymbol{\theta})$ |
|---|---|---|
| $\boldsymbol{\theta}_A$ | 0.90 | 0.85 |
| $\boldsymbol{\theta}_B$ | 0.35 | 0.80 |

The average scores are:

$$T_1 = 0.625, \quad T_2 = 0.825, \quad T_2 - T_1 = 0.20$$

The deviation advantages are:

$$\delta_{12} = \tfrac{1}{2}(+0.90 - 0.35) = 0.275, \quad \delta_{21} = \tfrac{1}{2}(-0.85 + 0.80) = -0.025$$

The differentiation condition $-\delta_{12} \leq T_1 - T_2 \leq \delta_{21}$ becomes:

$$-0.275 \leq -0.20 \leq -0.025$$

which holds. Hence, by Lemma C.2, the equilibrium is **full differentiated**: the two platforms select different models, even though $T_1 < T_2$.

The payoff matric of this scenario is:

| $\boldsymbol{f}$ | $g_1$ | $g_2$ |
|---|---|---|
| $g_1$ | $(0.3125, 0.3125)$ | $(0.45, 0.4)$ |
| $g_2$ | $(0.4, 0.45)$ | $(0.4125, 0.4125)$ |

So the equilibrium is $(g_1, g_2)$ or $(g_2, g_1)$.

**Scenario B:** We keep $T_1 = 0.625$, $T_2 = 0.825$, and $T_2 - T_1 = 0.20$, but change the type-level structure to weaken $g_1$'s advantage:

| | $S_1(\boldsymbol{\theta})$ | $S_2(\boldsymbol{\theta})$ |
|---|---|---|
| $\boldsymbol{\theta}_A$ | 0.60 | 0.70 |
| $\boldsymbol{\theta}_B$ | 0.65 | 0.95 |

The deviation advantages are now:
$$\delta_{12} = \tfrac{1}{2}(-0.60 - 0.65) = -0.625, \quad \delta_{21} = \tfrac{1}{2}(+0.70 + 0.95) = 0.825$$
The differentiation condition $-\delta_{12} \leq T_1 - T_2 \leq \delta_{21}$ becomes:
$$0.625 \leq -0.20 \leq 0.825$$
which fails. The consolidation condition $T_2 - T_1 > \delta_{12}$ or $T_1 - T_2 > \delta_{21}$ holds since $0.20 > -0.625$; thus, the equilibrium is **homogeneous** on $g_2$.

The payoff matric of this scenario is:

| $\boldsymbol{f}$ | $g_1$ | $g_2$ |
|---|---|---|
| $g_1$ | $(0.3125, 0.3125)$ | $(0, 0.825)$ |
| $g_2$ | $(0.825, 0)$ | $(0.4125, 0.4125)$ |

So the equilibrium is $(g_2, g_2)$.

### C.5 THE PROOF OF COROLLARY 3.5

**Corollary 3.5** (High User Centralization $\Rightarrow$ Homogeneous Equilibrium). *Assume there exists a dominant user type $\boldsymbol{\theta}^\star$ with fraction $\pi_{\boldsymbol{\theta}}^\star$ and a model $m$ satisfying: $\forall j \neq m$, $S_m(\boldsymbol{\theta}^\star) - S_j(\boldsymbol{\theta}^\star) \geq \rho > 0$ and $\forall \boldsymbol{\theta} \neq \boldsymbol{\theta}^\star, j \neq m$, $|S_j(\boldsymbol{\theta}) - S_m(\boldsymbol{\theta})| \leq \Gamma$. If $\pi_{\boldsymbol{\theta}}^\star$ is sufficiently large and satisfies*
$$\pi_{\boldsymbol{\theta}}^\star \geq 1 - \frac{1}{1 + 2\frac{\Gamma}{\rho}}$$
*then the homogeneous strategy $\boldsymbol{f}^* = (m, \ldots, m)$ is a pure-strategy Nash equilibrium.*

*Proof.* We use the utility decomposition Proposition 3.3
$$N \cdot U_i(\boldsymbol{f}) = T_{f_i} + \delta_{f_i}(\boldsymbol{f})$$

Suppose all players currently choose $m$, consider a deviation by a single platform to some $k \neq m$. Let $\Delta$ denote the utility gain from this deviation, then
$$\Delta = [T_k + \delta_k(\boldsymbol{f}_{-i}^* \cup \{k\})] - [T_m + \delta_m(\boldsymbol{f}^*)]$$
The $\Delta$ is consists of three parts:

**Loss on the dominant type $\boldsymbol{\theta}^\star$:** Under $\boldsymbol{f}^*$, each platform receives a $\frac{1}{N}$ share of $\boldsymbol{\theta}^\star$'s contribution $\frac{1}{N} S_m(\boldsymbol{\theta}^\star)$. After deviating to $k$, the deviator's share on $\boldsymbol{\theta}^\star$ becomes $0$ because $m$ strictly wins there. Using the margin $S_m(\boldsymbol{\theta}^\star) - S_k(\boldsymbol{\theta}^\star) \geq \rho$, the utility loss from $\boldsymbol{\theta}^\star$ is at least
$$\Delta U_{\boldsymbol{\theta}^\star} \geq \frac{\pi_{\boldsymbol{\theta}}^\star \rho}{N}$$

**Gain on minority types where $k$ wins:** On $\Theta \setminus \{\boldsymbol{\theta}^\star\}$, the total mass is $1 - \pi_{\boldsymbol{\theta}}^\star$. Wherever $k$ wins $m$, the deviator's share improves from $\frac{1}{N}$ to $1$. Since $k$'s per-type advantage over $m$ is at most $\Gamma$, the upper bound gain is
$$\Delta U_{\text{win}} \leq \frac{(1 - \pi_{\boldsymbol{\theta}}^\star)\Gamma}{N}$$

**Additional loss on minority types where $k$ loses:** On those types where $m$ remains superior, the deviator's share falls from $\frac{1}{N}$ to $0$. Bounding score levels by the same heterogeneity constant $\Gamma$, we get
$$\Delta U_{\text{lose}} \leq -\frac{(1 - \pi_{\boldsymbol{\theta}}^\star)\Gamma}{N}$$

So the change:
$$\begin{aligned} \Delta &= \Delta U_{\text{win}} - \Delta U_{\text{lose}} - \Delta U_{\boldsymbol{\theta}^\star} \\ &= \frac{(1 - \pi_{\boldsymbol{\theta}}^\star)\Gamma}{N} + \frac{(1 - \pi_{\boldsymbol{\theta}}^\star)\Gamma}{N} - \frac{\pi_{\boldsymbol{\theta}}^\star \rho}{N} \\ &= \frac{2(1 - \pi_{\boldsymbol{\theta}}^\star)\Gamma - \pi_{\boldsymbol{\theta}}^\star \rho}{N} \end{aligned}$$

Therefore, if $\Delta \leq 0$, $2(1 - \pi_{\boldsymbol{\theta}}^{\star})\Gamma - \pi_{\boldsymbol{\theta}}^{\star}\rho \leq 0$, that is:

$$\pi_{\boldsymbol{\theta}}^{\star} \geq 1 - \frac{\rho}{\rho + 2\Gamma}$$

so no player benefits from deviating and the homogeneous profile $\boldsymbol{f}^{*}$ is a Nash equilibrium. So the condition is:

$$\boxed{\pi_{\boldsymbol{\theta}}^{\star} \geq 1 - \frac{\rho}{\rho + 2\Gamma}}$$

$\square$

## C.6 PROOF OF PROPOSITION 4.1

**Proposition C.3.** *Consider the case where $N = 2$ and fix a strategy $\boldsymbol{f} = (g_i, g_j)$. For each user type $\boldsymbol{\theta} \in \Theta$ with weight $\pi(\boldsymbol{\theta}) \geq 0$, the coverage value of the pair $(i, j)$ is*

$$V(i, j) = \frac{1}{2}\left(T_i + T_j + \delta_{ij} + \delta_{ji}\right)$$

*Proof.* First, for any $i \neq j$,

$$\delta_{ij} + \delta_{ji} = \sum_{\boldsymbol{\theta} \in \Theta} \pi_{\boldsymbol{\theta}} |S_i(\boldsymbol{\theta}) - S_j(\boldsymbol{\theta})| \tag{18}$$

Fix a type $\boldsymbol{\theta}$. Consider three cases.

**Case 1:** $S_i(\boldsymbol{\theta}) > S_j(\boldsymbol{\theta})$: Then $Z_{ij}(\boldsymbol{\theta}) = S_i(\boldsymbol{\theta})$ and $Z_{ji}(\boldsymbol{\theta}) = -S_j(\boldsymbol{\theta})$, so $Z_{ij}(\boldsymbol{\theta}) + Z_{ji}(\boldsymbol{\theta}) = S_i(\boldsymbol{\theta}) - S_j(\boldsymbol{\theta}) = |S_i(\boldsymbol{\theta}) - S_j(\boldsymbol{\theta})|$.

**Case 2:** $S_i(\boldsymbol{\theta}) < S_j(\boldsymbol{\theta})$: Then $Z_{ij}(\boldsymbol{\theta}) = -S_i(\boldsymbol{\theta})$ and $Z_{ji}(\boldsymbol{\theta}) = S_j(\boldsymbol{\theta})$, so $Z_{ij}(\boldsymbol{\theta}) + Z_{ji}(\boldsymbol{\theta}) = S_j(\boldsymbol{\theta}) - S_i(\boldsymbol{\theta}) = |S_i(\boldsymbol{\theta}) - S_j(\boldsymbol{\theta})|$.

**Case 3:** $S_i(\boldsymbol{\theta}) = S_j(\boldsymbol{\theta})$: Then $Z_{ij}(\boldsymbol{\theta}) = Z_{ji}(\boldsymbol{\theta}) = 0$, hence the sum is $0 = |S_i(\boldsymbol{\theta}) - S_j(\boldsymbol{\theta})|$.

Multiplying by $\pi_{\boldsymbol{\theta}}$ and summing over $\boldsymbol{\theta}$ yields the claim.

Use the pointwise identity: $\max\{a, b\} = \frac{1}{2}(a + b + |a - b|)$. Applying it with $a = S_i(\boldsymbol{\theta})$ and $b = S_j(\boldsymbol{\theta})$ and summing over $\boldsymbol{\theta}$:

$$
\begin{aligned}
V(i, j) &= \sum_{\boldsymbol{\theta}} \pi(\boldsymbol{\theta}) \max\{S_i(\boldsymbol{\theta}), S_j(\boldsymbol{\theta})\} \\
&= \frac{1}{2} \sum_{\boldsymbol{\theta}} \pi(\boldsymbol{\theta})\left(S_i(\boldsymbol{\theta}) + S_j(\boldsymbol{\theta}) + |S_i(\boldsymbol{\theta}) - S_j(\boldsymbol{\theta})|\right) \\
&= \frac{1}{2}\left(\underbrace{\sum_{\boldsymbol{\theta}} \pi(\boldsymbol{\theta})S_i(\boldsymbol{\theta})}_{T_i} + \underbrace{\sum_{\boldsymbol{\theta}} \pi(\boldsymbol{\theta})S_j(\boldsymbol{\theta})}_{T_j} + \underbrace{\sum_{\boldsymbol{\theta}} \pi(\boldsymbol{\theta})|S_i(\boldsymbol{\theta}) - S_j(\boldsymbol{\theta})|}_{\delta_{ij}+\delta_{ji}\text{by Eq. 18}}\right) \\
&= \frac{1}{2}\left(T_i + T_j + \delta_{ij} + \delta_{ji}\right)
\end{aligned}
$$

$\square$

**Proposition 4.1.** *[Coverage Value Calculation]  Given a strategy profile $\boldsymbol{f} = (f_1, \ldots, f_N)$, the coverage value in Definition 2.4 can be written as:*

$$V(\boldsymbol{f}) = \frac{1}{N} \sum_{i=1}^{N} \left(T_{f_i} + \delta_{f_i}(\boldsymbol{f})\right)$$

*where $T$ and $\delta$ are defined in Definitions 3.1 and 3.2, respectively.*

*Proof.* Fix $\boldsymbol{\theta}$ and recall $\mathbb{A}_{\boldsymbol{f}}(\boldsymbol{\theta}) := \arg\max_{k \in \mathbb{M}(\boldsymbol{f})} S_k(\boldsymbol{\theta})$ and $A_{\boldsymbol{f}}(\boldsymbol{\theta}) = |\mathbb{A}_{\boldsymbol{f}}(\boldsymbol{\theta})|$. By the definition of $Z_j(\boldsymbol{\theta}; \boldsymbol{f})$:

$$\sum_{j \in \mathbb{M}(\boldsymbol{f})} (S_j(\boldsymbol{\theta}) + Z_j(\boldsymbol{\theta}; \boldsymbol{f})) = \sum_{j \in \mathbb{A}_{\boldsymbol{f}}(\boldsymbol{\theta})} \left(1 + \frac{N - A_{\boldsymbol{f}}(\boldsymbol{\theta})}{A_{\boldsymbol{f}}(\boldsymbol{\theta})}\right) S_j(\boldsymbol{\theta})$$

$$= \frac{N}{A_{\boldsymbol{f}}(\boldsymbol{\theta})} \sum_{j \in \mathbb{A}_{\boldsymbol{f}}(\boldsymbol{\theta})} S_j(\boldsymbol{\theta})$$

$$= N \max_{k \in \mathbb{M}(\boldsymbol{f})} S_k(\boldsymbol{\theta})$$

Multiply by $\pi(\boldsymbol{\theta})$, sum over $\boldsymbol{\theta}$, and divide by $N$ to obtain

$$V(\boldsymbol{f}) = \frac{1}{N} \sum_{j \in \mathbb{M}(\boldsymbol{f})} \left(\sum_{\theta} \pi(\boldsymbol{\theta}) S_j(\boldsymbol{\theta}) + \sum_{\theta} \pi(\boldsymbol{\theta}) Z_j(\boldsymbol{\theta}; \boldsymbol{f})\right)$$

$$= \frac{1}{N} \sum_{j \in \mathbb{M}(\boldsymbol{f})} (T_j + \delta_j(\boldsymbol{f}))$$

$$= \frac{1}{N} \sum_{i=1}^{N} (T_{f_i} + \delta_{f_i}(\boldsymbol{f}))$$

$\square$

## C.7 PROOF OF LEMMA 4.2

**Lemma 4.2.** *Let $W$ denote the user welfare (Definition 2.5) achieved under the game $\mathcal{G}(\mathbb{G}, \mathbb{I}, \Theta)$, and $W_{\mathrm{opt}}$ the social optimum welfare (Definition 2.6). Then, it always holds that $W \leq W_{\mathrm{opt}}$.*

*Proof.* If $\mathcal{O} = \boldsymbol{f}^*$ is a PNE, then $W(\mathcal{O}) = V(\boldsymbol{f}^*) \leq \max_{\boldsymbol{f}} V(\boldsymbol{f}) = W_{\mathrm{opt}}$ by definition.

If $\mathcal{O}$ is a cycle $\boldsymbol{f}^{(1)}, \ldots, \boldsymbol{f}^{(L)}$, then

$$W(\mathcal{O}) = \frac{1}{L} \sum_{t=1}^{L} V(\boldsymbol{f}^{(t)}) \leq \frac{1}{L} \sum_{t=1}^{L} \max_{\boldsymbol{f}} V(\boldsymbol{f}) = W_{\mathrm{opt}}$$

since an arithmetic mean is at most its maximum term.

Therefore, $\boxed{W(\mathcal{O}) \geq W_{\mathrm{opt}}}$ $\square$

## C.8 THE EXAMPLE OF LEMMA 4.2

*Example* C.4. Consider three user types $\boldsymbol{\theta}_A, \boldsymbol{\theta}_B, \boldsymbol{\theta}_C$ with weights $\pi(\boldsymbol{\theta}_A) = 0.5$, $\pi(\boldsymbol{\theta}_B) = 0.3$, $\pi(\boldsymbol{\theta}_C) = 0.2$. Their scores for each of the three models $g_1, g_2, g_3$ are:

| | $g_1$ | $g_2$ | $g_3$ |
|---|---|---|---|
| $\boldsymbol{\theta}_A$ | 0.434 | 0.698 | 0.760 |
| $\boldsymbol{\theta}_B$ | 0.828 | 0.679 | 0.431 |
| $\boldsymbol{\theta}_C$ | 0.343 | 0.776 | 0.565 |

The average scores are:

$$T_1 = 0.534, \quad T_2 = 0.7079, \quad T_3 = 0.6223.$$

The pairwise attraction shifts $\delta_{ij}$ are computed as in the model:

$$\delta_{12} = -0.0372, \quad \delta_{21} = 0.3005, \quad \delta_{13} = -0.0372$$

$$\delta_{31} = 0.3637, \quad \delta_{23} = 0.0099, \quad \delta_{32} = 0.1377$$

The coverage value of a pair $(g_i, g_j)$ is

$$V(i, j) = \sum_\theta \pi_\theta \max\{S_\theta(g_i), S_\theta(g_j)\}$$

Numerically:

$$V(1, 2) = 0.7526, \quad V(1, 3) = 0.7414, \quad V(2, 3) = 0.7389$$

Thus, the socially optimal pair is $(g_1, g_2)$ with

$$W_{\text{opt}} = 0.7526.$$

The payoff:

| $\boldsymbol{f}$ | $g_1$ | $g_2$ | $g_3$ |
|---|---|---|---|
| $g_1$ | $(0.267, 0.267)$ | $(0.2484, 0.5042)$ | $(0.2484, 0.5112)$ |
| $g_2$ | $(0.5042, 0.2484)$ | $(0.35395, 0.35395)$ | $(0.3589, 0.38)$ |
| $g_3$ | $(0.5112, 0.2484)$ | $(0.38, 0.3589)$ | $(0.31115, 0.31115)$ |

Equilibrium check for $(g_2, g_3)$: the differentiation condition requires:

$$\begin{cases} T_2 + \delta_{23} \geq \max\{T_3, \, T_1 + \delta_{13}\} \\ T_3 + \delta_{32} \geq \max\{T_2, \, T_1 + \delta_{12}\} \end{cases}$$

Substituting:

$$T_2 + \delta_{23} = 0.7079 + 0.0099 = 0.7178$$
$$\max\{T_3, \, T_1 + \delta_{13}\} = \max\{0.6223, \, 0.534 - 0.0372\} = 0.6223$$
$$T_3 + \delta_{32} = 0.6223 + 0.1377 = 0.7600$$
$$\max\{T_2, \, T_1 + \delta_{12}\} = \max\{0.7079, \, 0.534 - 0.0372\} = 0.7079$$

Both inequalities hold, hence $(g_2, g_3)$ is a full differentiated equilibrium with welfare

$$W_{\text{eq}} = V(2, 3) = 0.7389.$$

Although $(g_2, g_3)$ is a valid differentiated equilibrium, it yields lower welfare than the optimal pair $(g_1, g_2)$:

$$\boxed{W_{\text{eq}} = 0.7389 \; < \; W_{\text{opt}} = 0.7526}$$

This demonstrates that a differentiated equilibrium does not necessarily coincide with socially optimal differentiation.

## C.9 PROOF OF PROPOSITION 4.3

**Proposition 4.3.** *Consider a game $\mathcal{G}(\mathbb{G}, \mathbb{I}, \Theta)$ with an equilibrium $\boldsymbol{f}^*$. Let $\widehat{\mathcal{G}}(\mathbb{G}, \mathbb{I}' := \mathbb{I} \cup \{i^+\}, \Theta)$ be another game with one additional platform added. Suppose there exists a model $h \in \mathbb{G}$ and an incumbent equilibrium strategy $\widehat{\boldsymbol{f}}$ from $\boldsymbol{f}^*$ such that the extended profile $\widehat{\boldsymbol{f}} := (\boldsymbol{f}^*, h)$ satisfies the best-response conditions: (i) the best response to $\boldsymbol{f}^*$ is $h$; (ii) no incumbent platform has a profitable deviation against $\widehat{\boldsymbol{f}}$. Then $\widehat{\boldsymbol{f}}$ is an equilibrium of the game $\widehat{\mathcal{G}}$. Furthermore, the user welfare and market diversity in $\widehat{\mathcal{G}}$ are at least as high as in $\mathcal{G}$, i.e., $\widehat{W} \geq W$ and $\widehat{D}_{\text{supp}} \geq D_{\text{supp}}$.*

*Proof.* Best-response conditions (i) and (ii) imply $\hat{\boldsymbol{f}}$ is a PNE. If $\hat{\boldsymbol{f}}$ belongs to a cycle, appending $h$ yields a one-step extension that meets the same no-improvement conditions for that period, so the induced outcome is an equilibrium.

For welfare, by the Proposition 4.1, since $\mathbb{M}(\hat{\boldsymbol{f}}) = \mathbb{M}(\boldsymbol{f}^*) \cup \{h\}$, for every type $\boldsymbol{\theta}$ we have

$$\max_{k \in \mathbb{M}(\hat{\boldsymbol{f}})} S_k(\boldsymbol{\theta}) \geq \max_{k \in \mathbb{M}(\boldsymbol{f}^*)} S_k(\boldsymbol{\theta})$$

Summing with weights $\pi(\boldsymbol{\theta})$: $V(\hat{\boldsymbol{f}}) \geq V(\boldsymbol{f}^*)$.

If $h \notin \mathbb{M}(\boldsymbol{f}^*)$ and improves some type strictly, then the inequality is strict. $\square$

*Example* C.5 (counterexample: two $\rightharpoonup$ three models). Two user types $\Theta = \{\boldsymbol{\theta}_A, \boldsymbol{\theta}_B\}$ with equal weights $\pi(\boldsymbol{\theta}_A) = \pi(\boldsymbol{\theta}_B) = 0.5$.

**Scenario A:** The model scores are:

| | $S_1(\boldsymbol{\theta})$ | $S_2(\boldsymbol{\theta})$ |
|---|---|---|
| $\boldsymbol{\theta}_A$ | 0.90 | 0.85 |
| $\boldsymbol{\theta}_B$ | 0.35 | 0.80 |

The average scores are:

$$T_1 = 0.625, \quad T_2 = 0.825$$

The deviation advantages are:

$$\delta_{12} = \tfrac{1}{2}(+0.90 - 0.35) = 0.275, \quad \delta_{21} = \tfrac{1}{2}(-0.85 + 0.80) = -0.025$$

The payoff matric of this scenario is:

| $f$ | $g_1$ | $g_2$ |
|---|---|---|
| $g_1$ | $(0.3125, 0.3125)$ | $(0.45, 0.4)$ |
| $g_2$ | $(0.4, 0.45)$ | $(0.4125, 0.4125)$ |

So the equilibrium is $(g_1, g_2)$ or $(g_2, g_1)$, and $W(\mathcal{O}) = V(1,2) = 0.85$

**Scenario B:** Add a new model $g_3$ with $S_3(\boldsymbol{\theta}_A) = 0.91, S_3(\boldsymbol{\theta}_B) = 0.77$ Then $T_3 = 0.84$ The deviation advantages are now:

$$\delta_{12} = \tfrac{1}{2}(+0.90-0.35) = 0.275, \quad \delta_{21} = \tfrac{1}{2}(-0.85+0.80) = -0.025 \quad \delta_{13} = \tfrac{1}{2}(-0.90-0.35) = -0.625$$

$$\delta_{31} = \tfrac{1}{2}(+0.91+0.77) = 1.68, \quad \delta_{23} = \tfrac{1}{2}(-0.85+0.80) = -0.025, \quad \delta_{32} = \tfrac{1}{2}(+0.91-0.77) = 0.07$$

The payoff matric of this scenario is:

| $f$ | $g_1$ | $g_2$ | $g_3$ |
|---|---|---|---|
| $g_1$ | $(0.3125, 0.3125)$ | $(0.45, 0.4)$ | $(0, 0.84)$ |
| $g_2$ | $(0.45, 0.4)$ | $(0.4125, 0.4125)$ | $(0.4, 0.455)$ |
| $g_3$ | $(0.84, 0)$ | $(0.455, 0.4)$ | $(0.42, 0.42)$ |

So the equilibrium is $(g_3, g_3)$, and $W(\mathcal{O}) = V(3,3) = 0.84$.

Here, $0.84 < 0.85$

*Example* C.6 (counterexample: two $\rightharpoonup$ three players). Consider user types $\pi(\boldsymbol{\theta}_A) = 0.18$, $\pi(\boldsymbol{\theta}_B) = 0.17$, $\pi(\boldsymbol{\theta}_C) = 0.16$, $\pi(\boldsymbol{\theta}_D) = 0.16$, $\pi(\boldsymbol{\theta}_E) = 0.17$, $\pi(\boldsymbol{\theta}_F) = 0.16$

The model scores are:

| | $S_1(\boldsymbol{\theta})$ | $S_2(\boldsymbol{\theta})$ | $S_3(\boldsymbol{\theta})$ | $S_4(\boldsymbol{\theta})$ | $S_5(\boldsymbol{\theta})$ | $S_6(\boldsymbol{\theta})$ |
|---|---|---|---|---|---|---|
| $\boldsymbol{\theta}_A$ | 0.030658748 | 0.208093837 | **0.32744655** | 0.298774868 | 0.154842913 | 0.020151094 |
| $\boldsymbol{\theta}_B$ | 0.021978186 | 0.149636775 | **0.298145086** | 0.274754494 | 0.092761844 | 0.014372437 |
| $\boldsymbol{\theta}_C$ | **0.266589463** | 0.035725005 | 0.019578686 | 0.029395873 | 0.04788997 | 0.182804301 |
| $\boldsymbol{\theta}_D$ | **0.171553999** | 0.007992042 | 0.007932614 | 0.019235272 | 0.067757338 | 0.160182327 |
| $\boldsymbol{\theta}_E$ | 0.039888468 | **0.145473659** | 0.077957489 | 0.078738138 | 0.110034101 | 0.019024562 |
| $\boldsymbol{\theta}_F$ | 0.131100401 | 0.089481771 | 0.136355415 | 0.132456332 | 0.095638528 | **0.136379898** |

**Scenario A:** With only two platform: the equilibrium is $(g_3, g_6)$ with user welfare $W \approx 0.2148$

**Scenario B:** With three platforms: the cycle is $(g_3, g_3, g_1) \rightarrow (g_3, g_3, g_6) \rightarrow (g_1, g_3, g_6) \rightarrow (g_1, g_3, g_3)$ and $W = (V(g_3, g_3, g_1) + V(g_3, g_3, g_6) + V(g_1, g_3, g_6))/3 \approx (0.2147 + 0.2148 + 0.199571)/3 = 0.2097$

Since $0.214 > 0.210$, adding a platform may not increase the user welfare.

C.10   EXTENSION TO SOFTMAX USER CHOICE MODEL

**Proposition C.7** (Robust nonexistence of PNE under softmax choice). *Consider a fixed instance* $(\Theta, \pi, \{S_j(\boldsymbol{\theta})\}_j)$. *Let* $U_i^{\mathrm{hard}}(\boldsymbol{f})$ *denote platform utilities under the hardmax user choice rule Eq. 1, and suppose that the induced platform game admits no pure Nash equilibrium, in the following strict sense: there exists* $\Delta > 0$ *such that for every profile* $\boldsymbol{f}$ *there is a platform* $i$ *and a deviation* $f_i'$ *with:*

$$U_i^{\mathrm{hard}}(f_i', \boldsymbol{f}_{-i}) \geq U_i^{\mathrm{hard}}(f_i, \boldsymbol{f}_{-i}) + \Delta \tag{19}$$

*Let* $U_i^{\mathrm{soft}}(f; \tau)$ *be the utilities under the softmax user choice rule Eq. 8 with* $\tau > 0$. *Then there exists* $\tau_0 > 0$ *such that for* $\forall\, 0 < \tau \leq \tau_0$, *the softmax game* $(U_i^{\mathrm{soft}}(\cdot; \tau))$ *also admits no pure Nash equilibrium.*

*Proof.* Fix a profile $\boldsymbol{f}$ and a type $\boldsymbol{\theta}$. Under hardmax, a type $\boldsymbol{\theta}$ user only considers platforms whose model achieves the highest score $\max_k S_{f_k}(\boldsymbol{\theta})$, assigns equal probability to those platforms, and assigns zero probability to all others. Under the softmax rule Eq. 8

$$p_i^{\mathrm{soft}}(\boldsymbol{\theta}) := \frac{e^{S_{f_i}(\boldsymbol{\theta})/\tau}}{\sum_{k=1}^N e^{S_{f_k}(\boldsymbol{\theta})/\tau}}$$

as $\tau \to 0$, the largest-score terms dominate the denominator, so $p_i^{\mathrm{soft}}(\boldsymbol{\theta}; \tau) \to p_i^{\mathrm{hard}}(\boldsymbol{\theta})$.

Platform utilities are finite weighted sums of these probabilities:

$$U_i^{\mathrm{soft}}(\boldsymbol{f}) = \sum_{\boldsymbol{\theta}} \pi_{\boldsymbol{\theta}} p_i^{\mathrm{soft}}(\boldsymbol{\theta}) S_{f_i}(\boldsymbol{\theta})$$

hence $U_i^{\mathrm{soft}}(\boldsymbol{f}) \to U_i^{\mathrm{hard}}(\boldsymbol{f})$ as $\tau \to 0$.

Because the strategy space is finite, this convergence is uniform over all profiles $f$: for any $\varepsilon > 0$ there exists $\tau_0 > 0$ such that for all $0 < \tau \leq \tau_0$, all platforms $i$, and all profiles $f$,

$$\left| U_i^{\mathrm{soft}}(\boldsymbol{f}) - U_i^{\mathrm{hard}}(\boldsymbol{f}) \right| \leq \varepsilon$$

Consider any profile $\boldsymbol{f}$. By the strict no NE condition Eq. 19, there exist $i$ and $f_i'$ with

$$U_i^{\mathrm{hard}}(f_i'; \boldsymbol{f}_{-i}) - U_i^{\mathrm{hard}}(f_i; \boldsymbol{f}_{-i}) \geq \Delta$$

Choose $\varepsilon = \Delta/4$ and the corresponding $\tau_0$. For any $0 < \tau \leq \tau_0$,

$$U_i^{\mathrm{soft}}(f_i'; \boldsymbol{f}_{-i}) - U_i^{\mathrm{soft}}(f_i; \boldsymbol{f}_{-i}) \geq \left[ U_i^{\mathrm{hard}}(f_i'; \boldsymbol{f}_{-i}) - \varepsilon \right] - \left[ U_i^{\mathrm{hard}}(f_i; \boldsymbol{f}_{-i}) + \varepsilon \right]$$
$$\geq \Delta - 2\varepsilon = \Delta/2 > 0$$

Thus $\boldsymbol{f}$ cannot be a best response strategy in the softmax user choice. Since $\boldsymbol{f}$ was arbitrary, the softmax game has no pure Nash equilibrium for any $0 < \tau \leq \tau_0$. $\square$

*Example* C.8. We provide a constructive counterexample here. Let $\Theta = \{\boldsymbol{\theta}_A, \boldsymbol{\theta}_B\}$ with uniform weights $\pi(\boldsymbol{\theta}_k) = 0.5$. Let $\mathbb{G} = \{g_1, g_2, g_3\}$ and define scores

|  | $S_1(\boldsymbol{\theta})$ | $S_2(\boldsymbol{\theta})$ | $S_3(\boldsymbol{\theta})$ |
|---|---|---|---|
| $\boldsymbol{\theta}_A$ | 0.734 | 0.148 | 0.934 |
| $\boldsymbol{\theta}_B$ | 0.833 | 0.935 | 0.534 |

If the softmax user choice is used with $\tau = 0.1$, then when there are two players, the payoff matrix is:

| $\boldsymbol{f} = (f_1, f_2)$ | $g_1$ | $g_2$ | $g_3$ |
|---|---|---|---|
| $g_1$ | $(0.39175, 0.39175)$ | $(0.47634, 0.34381)$ | $(0.43853, 0.42549)$ |
| $g_2$ | $(0.34381, 0.4763)$ | $(0.27075, 0.27075)$ | $(0.45843, 0.47210)$ |
| $g_3$ | $(0.42549, 0.43853)$ | $(0.47210, 0.45843)$ | $(0.36925, 0.36925)$ |

Here, the cycle is $(g_3, g_1), (g_3, g_2), (g_1, g_2), (g_1, g_3), (g_2, g_3), (g_2, g_1)$.

The $W_{opt} = 0.9345$, but $W = (0.8835 + 0.9345 + 0.835)/3 = 0.87067$

We now show that the $T + \delta$ decomposition extends to the softmax user choice rule in Eq. 8. We keep Definition 3.1 for the average score $T_f$ unchanged, and adapt the attraction term and deviation advantage as follows:

**Definition C.9** (Attraction Term and Deviation Advantage of softmax). For a strategy profile $\boldsymbol{f} = (f_1, \ldots, f_N)$ with $f_i \in \mathbb{M}$, the *attraction term* for $f_i$ in strategy $\boldsymbol{f}$ is defined as

$$Z_{f_i}^{\text{soft}}(\boldsymbol{\theta}; \boldsymbol{f}) := \frac{(N-1)e^{S_{f_i}(\boldsymbol{\theta})/\tau} - \sum_{k \neq i} e^{S_{f_k}(\boldsymbol{\theta})/\tau}}{\sum_{k=1}^{N} e^{S_{f_k}(\boldsymbol{\theta})/\tau}} S_{f_i}(\theta) \tag{20}$$

The *deviation advantage* for $f_i$ under strategy $\boldsymbol{f}$ is defined as

$$\delta_{f_i}^{\text{soft}}(\boldsymbol{f}) := \sum_{\boldsymbol{\theta} \in \Theta} \pi_{\boldsymbol{\theta}} \cdot Z_{f_i}^{\text{soft}}(\boldsymbol{\theta}; \boldsymbol{f}) \tag{21}$$

With this definition, the utility decomposition in Proposition 3.3 continues to hold under softmax choice:

$$U_i^{\text{soft}}(f_i; \boldsymbol{f}_{-i}) = \frac{1}{N}\left(T_{f_i} + \delta_{f_i}^{\text{soft}}(\boldsymbol{f})\right) \tag{22}$$

*Proof.* We use $e^{\boldsymbol{f}}$ to denote $\sum_{f_j \in \boldsymbol{f}} e^{S_{f_j}(\boldsymbol{\theta})/\tau}$.

We have:

$$U_i^{\text{soft}}(f_i, \boldsymbol{f}_{-i}) = \sum_{\boldsymbol{\theta} \in \Theta} \pi(\boldsymbol{\theta}) \frac{e^{S_{f_i}(\boldsymbol{\theta})/\tau}}{e^{\boldsymbol{f}}} S_{f_i}(\boldsymbol{\theta}) \tag{23}$$

$$T_{f_i} = \sum_{\boldsymbol{\theta} \in \Theta} \pi(\boldsymbol{\theta}) S_{f_i}(\boldsymbol{\theta}) = \sum_{\boldsymbol{\theta} \in \Theta} \pi(\boldsymbol{\theta}) \frac{e^{S_{f_i}(\boldsymbol{\theta})/\tau} + e^{\boldsymbol{f}_{-i}}}{e^{\boldsymbol{f}}} S_{f_i}(\boldsymbol{\theta}) \tag{24}$$

$$\delta_{f_i}^{\text{soft}}(\boldsymbol{f}) = \sum_{\boldsymbol{\theta} \in \Theta} \pi_{\boldsymbol{\theta}} \cdot Z_{f_i}^{\text{soft}}(\boldsymbol{\theta}; \boldsymbol{f}) = \sum_{\boldsymbol{\theta} \in \Theta} \pi_{\boldsymbol{\theta}} \frac{(N-1)e^{S_{f_i}(\boldsymbol{\theta})/\tau} - e^{\boldsymbol{f}_{-1}}}{e^{\boldsymbol{f}}} S_{f_i}(\theta) \tag{25}$$

It is clear that $U_i^{\text{soft}}(f_i, \boldsymbol{f}_{-i}) = \frac{1}{N}\left(T_{f_i} + \delta_{f_i}^{\text{soft}}(\boldsymbol{f})\right)$ □

## D  EXPERIMENTS WITH SYNTHETIC DATASET

Table 3: Simulation model pool.

| Model | $b_j$ | $\mu_{jr}$ | $A_{jr}$ | $\sigma_{jr}$ |
|---|---|---|---|---|
| 1 | 0.12 | (1.5, 0.0) | 0.90 | 1.20 |
| 2 | 0.05 | (0.0, 0.0) | 1.30 | 0.35 |
| 3 | 0.08 | (3.0, 0.0) | 1.00 | 0.50 |
| 4 | 0.06 | (0.0,0.0), (3.0,0.0) | 0.70, 0.70 | 0.70, 0.70 |
| 5 | 0.05 | (1.5, 0.6) | 1.00 | 0.40 |
| 6 | 0.05 | (1.5,-0.6) | 1.00 | 0.40 |

In this section, we design a controlled simulation environment to study equilibrium outcomes under different models and user populations.

**Generative Models.** We consider $M$ generative models $\mathbb{G} = \{g_j\}_{j=1}^{M}$, each parameterized as a Radial Basis Function (RBF) (Broomhead & Lowe, 1988) mixture:

$$g_j(x) = b_j + \sum_{r=1}^{R_j} A_{jr} \cdot \exp\left(-\frac{1}{2\sigma_{jr}^2}\|x - \mu_{jr}\|^2\right)$$

where $R_j$ is the number of kernels for model $j$, $\mu_{jr}$ is the center of the $r$-th kernel, $\sigma_{jr}$ is its width, $A_{jr}$ is its amplitude, and $b_j$ is a bias. Outputs are truncated to $[0, 1]$.

**User Distributions**  We represent the user by $\Theta = \{\boldsymbol{\theta}_k\}_{k=1}^K$, where $\boldsymbol{\theta} \in \mathbb{R}^d$ has distribution $\pi_{\boldsymbol{\theta}}$. The distribution $\pi_{\boldsymbol{\theta}}$ is derived by discretizing a Gaussian Mixture Model (GMM) with $Q$ components, where each component $q$ is parameterized by weight $w_q \geq 0$ with $\sum_q w_q = 1$, mean vector $\mu_q$, and covariance matrix $\Sigma_q$:

$$\pi(u) = \sum_{q=1}^{Q} w_q \, \mathcal{N}(u \mid \mu_q(u), \Sigma_q(u)).$$

continuous samples $u$ drawn from this GMM are then mapped to the nearest discrete type $\boldsymbol{\theta}_k$, This construction yields a finite user distribution $\pi(\theta)$ that serves as input to the equilibrium analysis. The variant user groups are constructed by shifting all component means along the $x$-axis: $\mu_q(u) \mapsto \mu_q(u) + (dx, 0)$ where $dx$ controls the degree of population shift or by adjust different weight $w_q$.

Table 4: User distribution parameters.

| Weight $w_q$ | Mean $\mu_q$ | Covariance $\Sigma_q$ |
|---|---|---|
| 0.6 | (0.0, 0.0) | [[0.25, 0.0], [0.0, 0.25]] |
| 0.4 | (3.0, 0.0) | [[0.25, 0.0], [0.0, 0.25]]) |

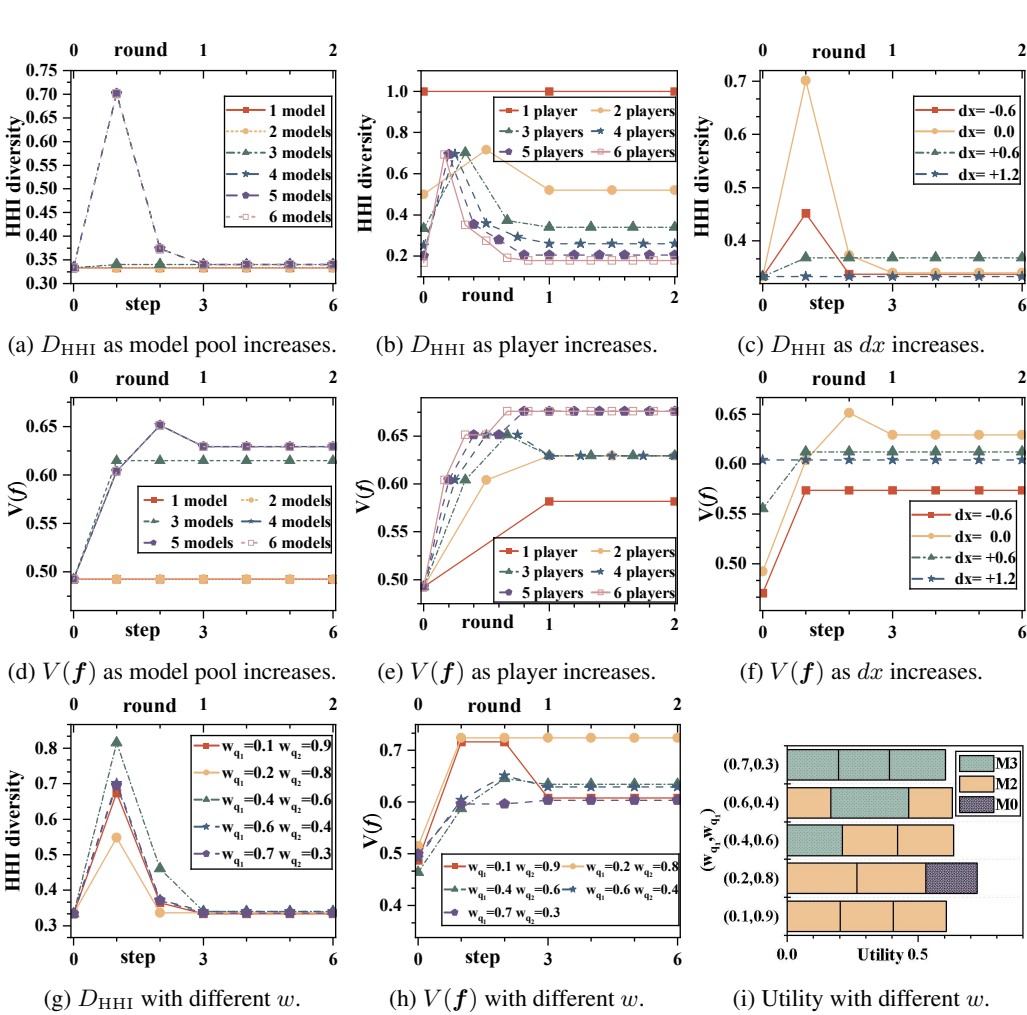

(a) $D_{\text{HHI}}$ as model pool increases.  (b) $D_{\text{HHI}}$ as player increases.  (c) $D_{\text{HHI}}$ as $dx$ increases.

(d) $V(\boldsymbol{f})$ as model pool increases.  (e) $V(\boldsymbol{f})$ as player increases.  (f) $V(\boldsymbol{f})$ as $dx$ increases.

(g) $D_{\text{HHI}}$ with different $w$.  (h) $V(\boldsymbol{f})$ with different $w$.  (i) Utility with different $w$.

Figure 7: Best-response simulations under different settings.

**Reward Function.**  The expected reward of model $j$ for user type $\theta$ is $S_j(\boldsymbol{\theta}) = \mathbb{E}_{x \sim g_j}[r_{\boldsymbol{\theta}}(x)]$. In theory, $g_j$ and $r_\theta$ are distinct objects, however, in our simulation, we collapse $g_j$ and $r_\theta$ into a single score function implemented as a radial basis function (RBF) mixture.

**Simulation Parameters and Results.** For all simulations, we have a model pool with $M = 6$ models as shown in Table 3 and $K = 12$ user types drawn from the GMMs. The baseline user distribution uses $Q = 2$ components as shown in Table 4.

We conduct four sets of experiments:

- Expanding the model pool. With three players fixed, we gradually enlarge the model pool size from 1 to 6. The resulting diversity $D_{\mathrm{HHI}}$ and coverage value $V(\boldsymbol{f})$ are shown in Fig. 7a and Fig. 7d, respectively.

- Increasing the number of players. With the full model pool available, we increase the number of players from 1 to 6. The resulting diversity $D_{\mathrm{HHI}}$ and coverage value $V(\boldsymbol{f})$ are shown in Fig. 7b and Fig. 7e, respectively.

- Shifting user groups. With the full model pool and three players, we vary the GMM means used to sample user types by setting $dx \in \{-0.6, 0.0, 0.6, 1.2\}$. The diversity $D_{\mathrm{HHI}}$ and coverage value $V(\boldsymbol{f})$ are shown in Fig. 7c and Fig. 7f, respectively.

- Changing mixture weights. With the full model pool and three players, we alter the GMM component weights $(w_{q_1}, w_{q_2}) \in \{(0.1, 0.9), (0.2, 0.8), (0.4, 0.6), (0.6, 0.4), (0.7, 0.3)\}$ The results are reported in Fig. 7g, Fig. 7h and Fig. 7i.

We observe that equilibria always exist in this setting, but diversity and welfare vary substantially depending on whether the new models are sufficiently differentiated. Strong but substitutable models lead to market homogenization, while genuinely differentiated entrants promote diversity and increase welfare.

## E  EXPERIMENTS WITH REAL DATASET

### E.1  DISCRETE BEST-RESPONSE SIMULATION

The models in the model pool are constructed by applying different LoRA parameters to the backbone network, each trained on different CIFAR-10 subsets, as summarized in Table. 1. The backbone network itself was trained on the full CIFAR-10 dataset for 200 epochs. During training, we used a learning rate of $2 \times 10^4$, 1000 diffusion steps, and a batch size of 256.

We first provide the average performance $T_i$ of the five models in model pool and their user-specific performance $S_i(\boldsymbol{\theta})$ in user groups in Fig. 8.

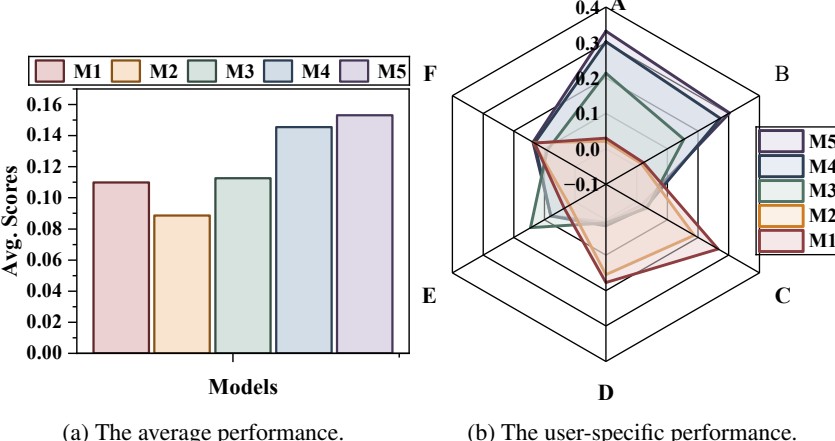

(a) The average performance.          (b) The user-specific performance.

Figure 8: The average performance $T_i$ of the five models in model pool in Table 1 and their user-specific performance $S_i(\boldsymbol{\theta})$ in Table 2.

**Impact of different user groups.** As a complementary experiment, we examine how platform choices vary when facing different user groups. Specifically, we consider three player choosing

Table 5: Different user pool

| User Pool | User type ($\theta$) with $\pi(\theta)$ |
|---|---|
| Pool 1 | A (0.2), B (0.2), E (0.2), F (0.2) |
| Pool 2 | A (0.3), C (0.3), E (0.4) |
| Pool 3 | C (0.2), D (0.2), E (0.35), F (0.33) |
| Pool 4 | A (0.6), F (0.4) |
| Pool 5 | A (0.1), B (0.09), C (0.16), D (0.32), E (0.17), F (0.16) |
| Pool 6 | C (0.33), D (0.35), E (0.2), F (0.12) |

Table 6: User type with its preference

| User type($\theta$) | Preferences(class($\theta_c$)) |
|---|---|
| A | cat (0.6), dog (0.4) |
| B | dog (0.7) cat (0.3) |
| C | airplane (0.5), ship (0.3), auto (0.2) |
| D | auto (0.6), truck (0.4) |
| E | bird (0.4), deer (0.3), frog (0.2), horse (0.1) |
| F | cat (0.2), dog (0.2), airplane (0.15), auto (0.15), ship (0.1), truck (0.2) |

Table 7: Outcomes of a 3-player setting under different user pools.

| User Pool | $f$ | $D_{\text{supp}}$ | $D_{\text{HHI}}$ | $W_{\text{eq}}$ |
|---|---|---|---|---|
| Pool 1 | $(M5, M5, M5)$ | 1 | 0.3333 | 0.2110 |
| Pool 2 | $(M5, M5, M1)$ | 2 | 0.3349 | 0.2117 |
| Pool 3 | $(M5, M3, M1)$ | 3 | 0.3335 | 0.1615 |
| Pool 4 | $(M5, M5, M5)$ | 1 | 0.3333 | 0.2366 |
| Pool 5 | $(M5, M1, M1)$ | 2 | 0.3856 | 0.1932 |
| Pool 6 | $(M1, M1, M1)$ | 1 | 0.3333 | 0.1715 |

from the model pool. The user group configurations are provided in Table. 5, and the corresponding equilibrium outcomes are summarized in Table. 7.

### E.2 ALGORITHMIC BEST-RESPONSE ENTRY

#### E.2.1 ALGORITHM DETAILS

We provide the implementation details and hyperparameters used in our experiments for evaluating algorithmic performance. We first describe the specific procedures of the Resampling and Direct-Gradient methods, followed by the hyperparameters employed in training and evaluation. Unless otherwise specified, the same base diffusion backbone and optimization settings are applied across methods for a fair comparison.

**Resampling.** The algorithm for resampling method is shown as Algorithm. 1.

---
**Algorithm 1** Training Data Resampling
---
**Require:** Dataset $\mathbb{D}$; user types $\Theta$; fixed opponents $\mathbb{G} = \{g_1, \cdots, g_M\}$ with scores $\{S_m(\theta)\}$; parameters $\beta, \gamma$; outer rounds $T$; inner epochs $E$; evaluation budget $b$.
1: Compute $\bar{S}(\theta) = \max_{j \in \mathbb{M}} S_j(\theta)$ for all $\theta$.
2: **for** $t = 1, \ldots, T$ **do**
3:     Estimate $S_\phi(\theta) := \mathbb{E}_{x_{1:b} \sim g_\phi} r_\theta(x)$.
4:     Compute $\Delta_\theta := S_\phi(\theta) - \bar{S}(\theta)$.
5:     Compute $\sigma_\theta = \sigma(\beta \Delta_\theta)$
6:     Type weights $\alpha_\theta = \pi(\theta)(\sigma_\theta)^\gamma \bar{S}(\theta)$.
7:     Data weights: $\hat{w}(u) \propto \sum_\theta \alpha_\theta q_\theta(u)$ or $\hat{w}(x) \propto \sum_\theta \alpha_\theta r_\theta(x)$.
8:     Sample $\mathbb{D}$ with $\hat{w}$ as $\hat{\mathbb{D}}$
9:     **for** $e = 1, \ldots, E$ **do**
10:         Update $\phi$ by minimizing the original loss in $\hat{\mathbb{D}}$.
11:     **end for**
12: **end for**
13: **return** $\phi$.

---

The specific parameter details for algorithm-level:

- Outer round, the time of resample $T = 5$.
- Inner epcohs $E = 50$.
- $\beta = 4$.
- $\gamma = 1$.
- Use adaptive scaling for $\sigma_{\boldsymbol{\theta}}$.
- LoRA rank 4, LoRA scaling coefficient 16, LoRA runtime multiplier 1.0.

**Direct-Gradient.** The algorithm for direct-gradient method is shown as Algorithm. 2.

---

**Algorithm 2** Direct-Gradient Optimization

---

**Require:** Dataset $\mathbb{D}$; user types $\Theta$; fixed opponents $\mathbb{G} = \{g_1, \cdots, g_M\}$ with scores $\{S_m(\boldsymbol{\theta})\}$; parameters $\lambda$; Epochs $E$;
1: **for** $e = 1, \ldots, E$ **do**
2:     Estimate $S_\phi(\boldsymbol{\theta}) := \mathbb{E}_{x_{1:b} \sim g_\phi} r_{\boldsymbol{\theta}}(x)$.
3:     Compute $\Delta_{\boldsymbol{\theta}} := S_\phi(\boldsymbol{\theta}) - \bar{S}(\boldsymbol{\theta})$.
4:     Compute $\sigma_{\boldsymbol{\theta}} = \sigma\left(\beta\Delta_{\boldsymbol{\theta}}\right)$
5:     $L \leftarrow l(\phi) - \lambda \sum_{\boldsymbol{\theta} \in \Theta} \pi(\boldsymbol{\theta}) \sigma_{\boldsymbol{\theta}} S_\phi(\boldsymbol{\theta})$
6:     $\phi \leftarrow \phi - \eta \nabla L.$
7: **end for**
8: **return** $\phi$

---

The specific parameter details for algorithm-level:

- Epcohs $E = 20$.
- $\lambda = 0.4$.
- Use adaptive scaling for $\sigma_{\boldsymbol{\theta}}$.

**Shared parameter.** The specific parameter details for shared training parameters:

- The backbone network is trained on the full CIFAR-10 dataset for 200 epochs.
- Batch size 256.
- Learning rate $2 \times 10^{-4}$ (for AdamW optimizer).
- Diffusion steps 1000.

**Data Distribution.** As a supplement to Fig. 6 A, we provide the label distributions of $2,000$ samples generated by three models, as shown in the Fig. 9.

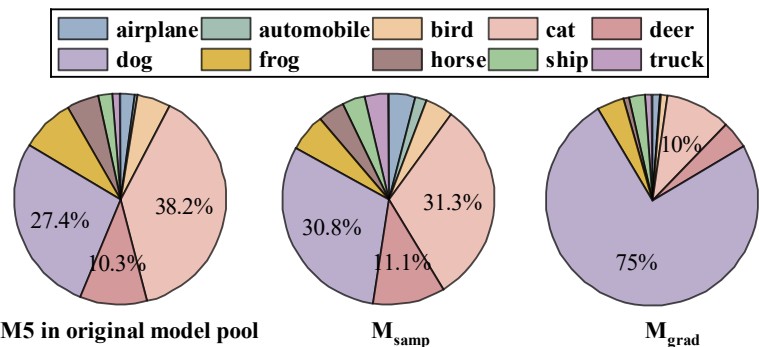

Figure 9: Label distributions of $2,000$ generated samples from three models: (a) left: $M2$ from the original model pool (b) middle: $M_{\mathrm{samp}}$ by redampling method. (c) right: $M_{\mathrm{grad}}$ by direct-gradient method.

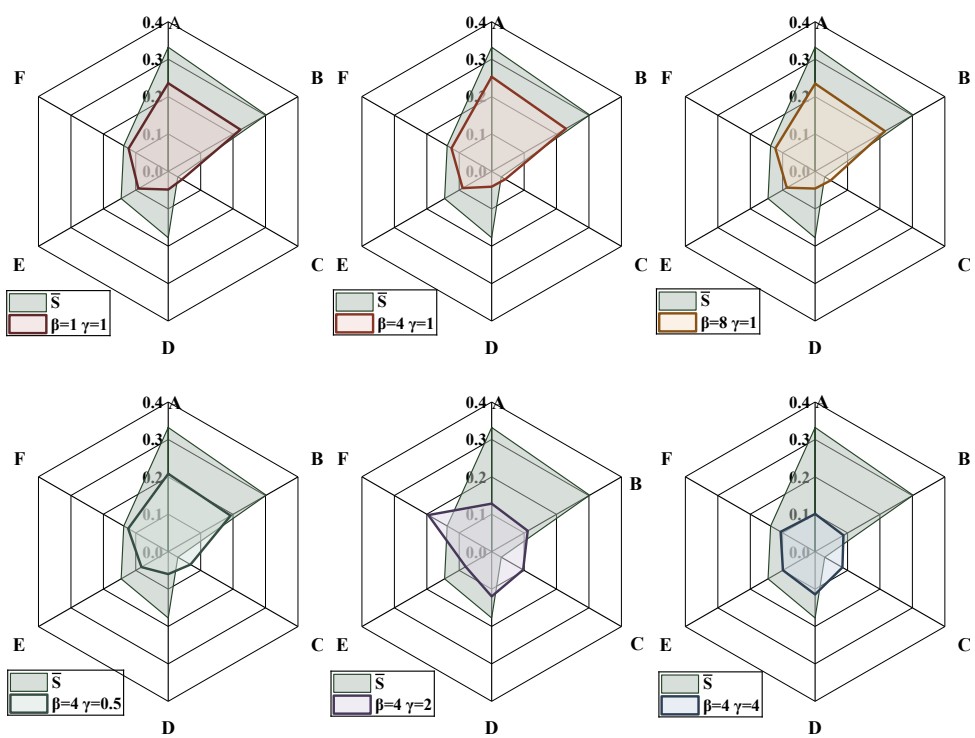

Figure 10: Different performance of model scores across user groups under different $\beta$ and $\gamma$ compared to each user group's best score $\bar{S}$.

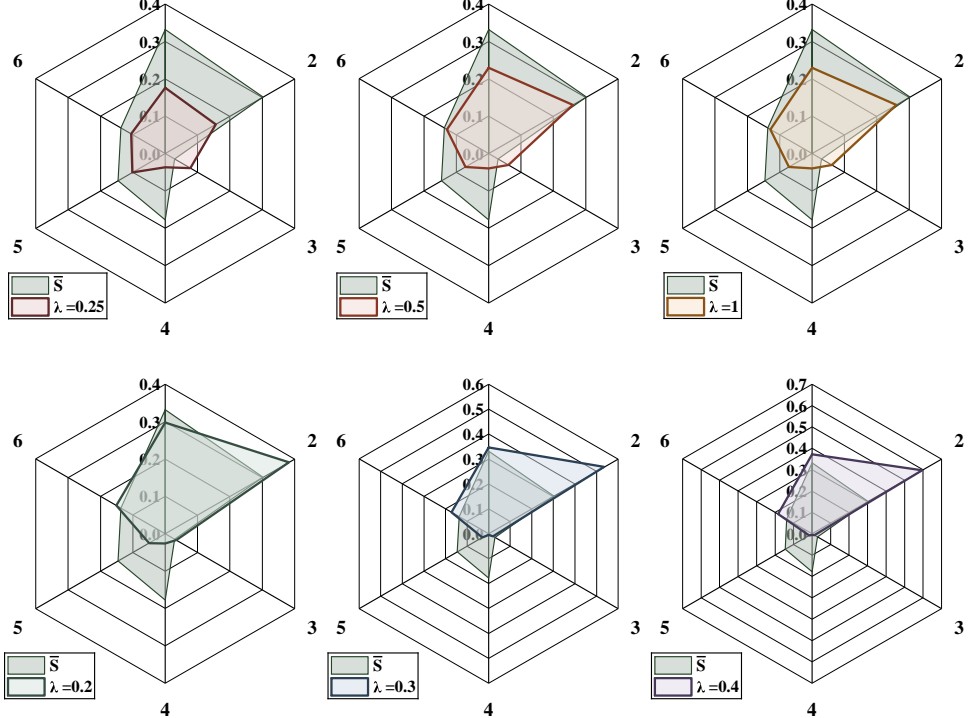

Figure 11: Different performance of model scores across user groups under different $\lambda$ compared to each user group's best score $\bar{S}$.

Table 8: Performance of four models on coding and reasoning benchmarks.

| Model | HEval | Multi-language | Overall | Math | IFEval |
|---|---|---|---|---|---|
| M0: CodeLlama-34B | 0.5079 | 0.4297 | 0.1721 | 0.0413 | 0.4604 |
| M1: Qwen2.5-Coder-32B | 0.5710 | 0.6497 | 0.3326 | 0.3089 | 0.4363 |
| M2: Nxcode-CQ-7B-orpo | 0.8723 | 0.6688 | 0.1237 | 0.4007 | 0.4007 |
| M3: Qwen2.5-Coder-32B-Instruct | 0.8320 | 0.7723 | 0.3989 | 0.4955 | 0.7265 |

Table 9: Different user pool

Table 10: User type with its preference

| User Pool | User type ($\theta$) with $\pi(\theta)$ |
|---|---|
| Pool 1 | A (0.2), B (0.2), C(0.2), D(0.2), E (0.2) |
| Pool 2 | A (0.1), B (0.1), C(0.2), D(0.5), E (0.1) |
| Pool 3 | A (0.35), B (0.2), C(0.2), D(0.35), E (0.2)) |

| User type($\theta$) | Preferences(class($\theta_c$)) |
|---|---|
| A | HEval (0.6), Overall (0.2), IFEval(0.2) |
| B | Overall (0.8), IFEval (0.2) |
| C | HEval (0.8), IFEval (0.2) |
| D | HEval (0.6), Muti-language (0.5) |
| E | Math (1.0) |

### E.2.2 ALGORITHM PARAMETER SENSITIVITY ANALYSIS

For resampling,we investigate the sensitivity of $\beta$ and $\gamma$. The result is shown as Fig. 10. For direct-gradient optimaiztion, We investigate the sensitivity of $\lambda$, which controls the trade-off between utility and the diffusion. The result is shown as Fig. 11.

## F EXPERIMENTS ON LANGUAGE MODELS

In this section, we study the three-layer game in a language setting using real large language models.

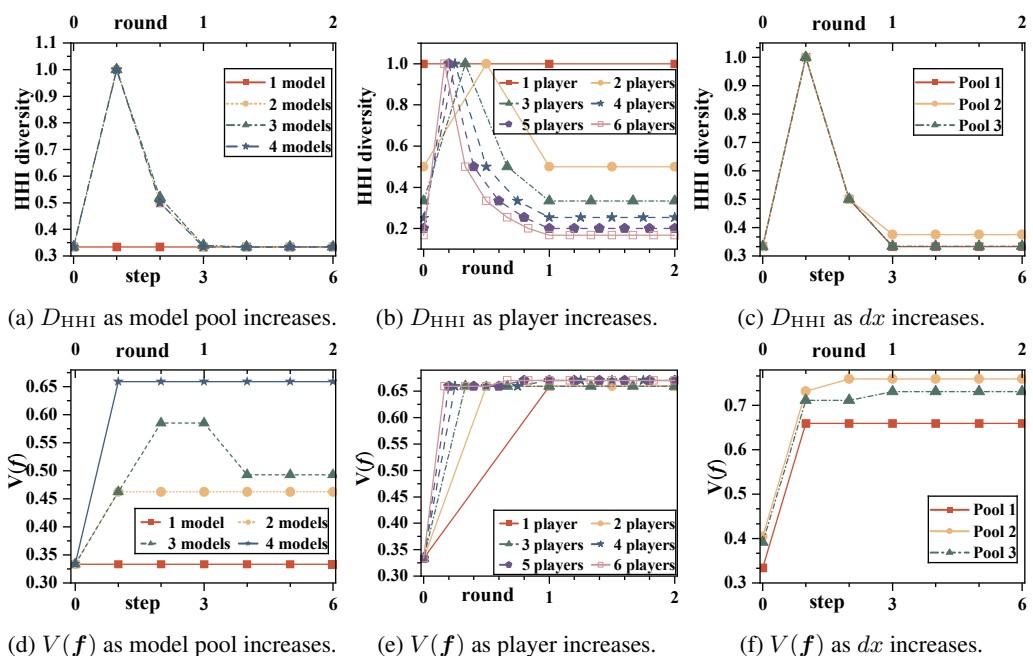

(a) $D_{\mathrm{HHI}}$ as model pool increases.  (b) $D_{\mathrm{HHI}}$ as player increases.  (c) $D_{\mathrm{HHI}}$ as $dx$ increases.

(d) $V(\boldsymbol{f})$ as model pool increases.  (e) $V(\boldsymbol{f})$ as player increases.  (f) $V(\boldsymbol{f})$ as $dx$ increases.

Figure 12: Best-response simulations under different settings of language tasks.

**Model Pool.** We consider a pool of four publicly available models that appear on both the Big-Code models leaderboard (BigCode, 2023) and the Open LLM Leaderboard (Fourrier et al., 2024): CodeLlama-34B (Rozière et al., 2024), Qwen2.5-Coder-32B (Hui et al., 2024), Nxcode-CQ-7B-

Table 11: Outcomes of a 3-player setting under different user pools.

| User Pool | $f$ | $D_{\text{supp}}$ | $D_{\text{HHI}}$ | $W_{\text{eq}}$ |
|-----------|-----|-------------------|------------------|-----------------|
| Pool 1 | $(M3, M3, M3)$ | 1 | 0.3333 | 0.65945 |
| Pool 2 | $(M3, M2, M2)$ | 2 | 0.375 | 0.75949 |
| Pool 3 | $(M3, M3, M2)$ | 2 | 0.33375 | 0.73080 |

orpo (Hong & Thorne, 2024) and Qwen2.5-Coder-32B-Instruct (Hui et al., 2024). For each model, we collect its HumanEval-Python score (HEval) and a multi-language coding score (Multi-language, average over Java, JavaScript, and C++) from the BigCode leaderboard, as well as its overall, math-related scores and instruction-following evaluation (IFEval) from the Open LLM Leaderboard. Table 8 summarizes these benchmark results, which we treat as pre-computed performance statistics.

**User Group and reward function.** We partition the user population into five groups, each characterized by heterogeneous preferences over metric preferences the details are given in Table 10. Then the reward for user type $\boldsymbol{\theta}$ is calculated by $r_{\boldsymbol{\theta}}(x) = \sum_{c \in \mathbb{C}} \theta_c \cdot \text{performance}$.

**Simulation and Results.** We conduct three sets of experiments:

- Expanding the model pool. With three players fixed, we gradually enlarge the model pool size from 1 to 4 face the user pool 1. The resulting diversity $D_{\text{HHI}}$ and coverage value $V(\boldsymbol{f})$ are shown in Fig. 12a and Fig. 12a, respectively.

- Increasing the number of players. With the full model pool available, we increase the number of players from 1 to 6 face the user pool 1. The resulting diversity $D_{\text{HHI}}$ and coverage value $V(\boldsymbol{f})$ are shown in Fig. 12b and Fig. 12e, respectively.

- Shifting user groups. With the full model pool and three players, we vary the user pool as shown in Table. 9 . The diversity $D_{\text{HHI}}$ and coverage value $V(\boldsymbol{f})$ are shown in Fig. 12c and Fig. 12f, respectively. The corresponding equilibrium outcomes are summarized in Table. 11.

## THE USE OF LARGE LANGUAGE MODELS

We used a large language model to aid in polishing grammar and phrasing. Consistent with ICLR policy, authors remain fully responsible for all content, including parts assisted by an LLM.

