# OpenReview forum: "Market Games for Generative Models: Equilibria, Welfare, and Strategic Entry"
_ICLR.cc/2026/Conference — ICLR 2026 Poster_

### Official Review · Reviewer_c5YV · 2025-10-28

**Soundness:** 3
**Presentation:** 3
**Contribution:** 3
**Rating:** 6
**Confidence:** 3

**Summary:**

This paper studies the game-theoretical formulation and its Nash equilibrium (NE) of the "model-platform-user" market exemplified by LLM applications. Specifically, the paper considers the following three-layered market: (1) LLL company releases models, (2) each service platform chooses which LLM to use in their service, and (3) users choose which LLMs to use for their purpose. The paper finds that NE has to conditions that each service uses a different model or all the services use the same model and highlights the finding that increasing the choice of model does not necessarily lead to a social optimum.

**Strengths:**

- The motivation for studying the "model-platform-user" market is well-explained, and I agree that this is a new and interesting market game to think about.

- The paper provides a solid game-theoretic analysis of the given problem, and the formulation seems reasonable for the most part.

- The paper also presents an algorithm to steer systems to achieve an equilibrium with diverse models, and shows that it works in the experiments.

**Weaknesses:**

- One discussion I have is whether the best-response dynamics are practical or not. In my understanding, the user and service greedily choose the best model or service for them. However, this may result in a quick change of behavior and does not seem realistic. Recent work on performative prediction considers a gradual change of participants (e.g., Brown et al., 22), and this looks like a more reasonable formulation.

- Another discussion is whether it is reasonable to consider a single reward. This is because services may focus on different tasks such as coding or translation, and depending on the application, what kind of aspects of LLM, may be different from each other. In such cases, I wonder if homogenization or "a-winner-takes-all" dynamics can really happen.

Brown et al., 22. Performative Prediction in a Stateful World. https://arxiv.org/abs/2011.03885

**Questions:**

- I didn't really understand why score Z can be the negative of S when f is not in A (Def 3.2). I understand that this is needed to show the utility decomposition in Prop 3.3., however, it is not quite intuitive why Z can be anti-proportional when f is not in A.

---

> ### Author Response · Authors · 2025-11-23
>
> We thank the reviewer for the constructive comments and for recognizing the motivation, formulation, and contributions of our work. We address the main concerns and questions below.
>
>
> > On the best-response dynamics. (W1)
>
> In the paper, best-response dynamics are used primarily as an ***analytical tool*** to study the **static game** and to determine whether a pure Nash equilibrium exists. They are not intended to model how users or platforms literally update their choices over time in real markets. Our main structural results (i.e., the existence and characterization of PNE) concern the equilibria of this static game, in which all platforms react simultaneously.
>
> We agree that real user and platform behavior is typically more gradual, with the data distribution shifting over repeated deployments and models being retrained iteratively. Performative prediction provides a natural framework for analyzing such gradual, stateful dynamics. However, incorporating these multi-round dynamics introduces substantially more complexity and is outside the scope of the present paper, which focuses on **first-round interactions**. Extending our three-layer framework to repeated adaptation and retraining is an important and exciting direction for future work.
>
> We have added the above discussion to the revised paper in Appendix B.
>
>
> > On a single reward. (W2)
>
> We thank the reviewer for raising this point. Our framework is designed for settings where platforms offer comparable services and draw from a shared pool of models $G$. In this setting, platforms face similar types of demand (e.g., overlapping mixes of coding and translation), and their strategic choice is which model from this common pool to deploy in order to attract different user groups. This is precisely the regime where direct competition and the possibility of homogenization or "winner-takes-all" outcomes are meaningful.
>
> By contrast, if different platforms specialize in largely disjoint services (e.g., one focuses exclusively on coding while another focuses exclusively on translation), then their effective model pools may overlap only partially or not at all. In that case, they are not competing for the shared user base in our sense (a user may use both services for translation and coding tasks, and platforms do not face a shared competitive environment). Such settings are better viewed as distinct sub-markets, for which our assumptions about a common model pool and overlapping user segments would not apply. Extending our framework to analyze these multi-market or partially overlapping-market environments is an interesting direction for future work.
>
> We have added the above discussion to the revised paper in Appendix B.
>
> > Z is anti-proportional when f is not in A. (Q)
>
> In Definition 3.2, $Z_{f_i}(\theta;f)$ measures relative advantage or disadvantage of each platform's model $f_i$ for a given user type $\theta$. Specifically, for platforms in the set of maximizers $A_f(\theta)$, $Z_{f_i}$ is positive and reflects the "attraction" or gain from being among the top-scoring platforms; for platforms not in the maximizer set, $Z_{f_i}(\theta;f)=-S_{f_i}(\theta)$ reflects the "opportunity loss" of not being chosen by this user type. This construction allows the deviation term $\delta_{f_i}(f) := \sum_{\theta \in \Theta} \pi_{\theta} \cdot Z_{f_i}(\theta;f)$ to capture a platform's net gain or loss relative to competitors in a linear, additive form.

---

### Official Review · Reviewer_ZFbA · 2025-10-31

**Soundness:** 3
**Presentation:** 3
**Contribution:** 3
**Rating:** 6
**Confidence:** 4

**Summary:**

This paper introduces a formal game-theoretic model of a three-layer generative AI market, comprising model providers, platforms, and heterogeneous users. It analyzes how platform competition for market share through model selection shapes equilibrium market structures, user welfare, and diversity. The key findings include necessary and sufficient conditions for the existence of pure Nash equilibria (both differentiated and homogeneous), a demonstration that increasing competition (via more models or platforms) does not necessarily improve welfare or diversity, and the proposal of two training methods for model providers to enter such a market strategically. Theoretical results are supported by experiments on synthetic and CIFAR-10 data.

**Strengths:**

	Originality: The three-layer model formulation is a timely and non-trivial extension of prior work, better capturing the structure of modern generative AI ecosystems.
	Theoretical Rigor: The analysis of equilibrium conditions, linking market structure to the balance between average performance and deviation advantage, is a solid theoretical contribution.
	Holistic Approach: The paper cohesively analyzes the market from platform, user, and (to a limited extent) model provider perspectives, providing a relatively comprehensive initial view.

**Weaknesses:**

	Strong and Potentially Unrealistic Assumptions: The model relies on several strong assumptions that limit the practical applicability of its conclusions.

Complete Information: The model assumes that platforms and model providers have perfect knowledge of the user type distribution π_θ and their reward functions r_θ(x). This ignores the significant challenge of preference learning and the strategic implications of information asymmetry in real-world markets.

Deterministic User Choice: The hard, winner-take-all user selection rule (Eq. 1) is a simplification. User behavior in practice is often stochastic, influenced by factors beyond a single quality score (e.g., UI, habit, discovery). A softer choice model (e.g., probabilistic) might lead to different equilibrium dynamics and welfare implications.

	Static Nature of Competition:

Non-Strategic Model Providers: The set of models G is treated as exogenous. Model providers are not modeled as strategic agents who dynamically develop or fine-tune models in response to market outcomes, which is a key feature of real-world competition.

Myopic Best-Response: The analysis of platform and entrant strategies primarily considers myopic best-response to static competitors. It does not fully capture the simultaneous, forward-looking strategic interactions where all agents anticipate and react to each other's moves in a multi-stage game, potentially altering equilibrium outcomes.

**Questions:**

	How would the central conclusions regarding equilibrium existence and market structure change if the deterministic user choice model were replaced with a stochastic one (e.g., a softmax function based on scores)? Could this alleviate the non-existence of PNE in some cases?

	The model relies on the strong assumption of complete information about user preferences. How do you expect the strategic dynamics and your conclusions to change in a more realistic setting with partial or asymmetric information, where platforms must learn user preferences over time?

	In the current framework, when platforms and entrants formulate their best responses, they treat competitors' strategies as static. If we adopt an equilibrium concept where all platforms "act simultaneously" and anticipate each other's reactions (such as Nash equilibrium, which relies on static assumptions in its analysis), do you believe the model entry strategy proposed in Section 5 would remain effective? How should a "strategic entrant," aware that its entry would trigger a recalibration of platform strategies, design its model?

---

> ### Author Response · Authors · 2025-11-23
>
> We thank the reviewer for the thoughtful and encouraging feedback, particularly the positive remarks on the originality of the three-layer model and the rigor and clarity of our analysis. We provide clarifications below to address the main concerns.
>
> > On Deterministic User Choice (W1, Q1)
>
> Indeed, the hardmax user choice model is a standard simplification commonly used in prior work on platform or model selection [1-3], where users deterministically select the platform with the highest score. To address the concern, we have extended our analysis to a softmax choice model and added a section (Section 5) in the revised paper. In this model, users select platforms probabilistically according to their scores (Eq.8), with a temperature parameter $\tau$ controlling the degree of stochasticity; the hardmax model is recovered as a special case when $\tau \to 0$. We clarify in Section 5 that:
>
> - The negative result on the existence of equilibrium (Proposition 2.2) is robust to the softmax extension. Any instance with no pure Nash equilibrium (PNE) under hardmax remains without a PNE for sufficiently small $\tau$ (Proposition C.7), and there exist instances with fixed $\tau>0$ where the softmax model also admits no PNE (Example C.8).
>
> - The utility decomposition (Proposition 3.3) and the equilibrium characterization for fully differentiated and homogeneous markets (Lemma 3.4) extend directly to the softmax case once the deviation term is redefined (Definition C.9). The equilibrium conditions retain the same form and can still be expressed as inequalities involving $T_{f_i}$ and $\delta^{\text{soft}}_{f_i}(f)$. This shows that our qualitative insights, such as the role of model performance heterogeneity in supporting differentiated equilibria and the misalignment between platform incentives and user welfare, hold beyond the hardmax assumption.
>
> - For welfare analysis, the coverage-based measure $V(f)$ depends only on which models are available and is therefore unaffected by the choice of hardmax or softmax. The only change is in the relation between coverage $V(f)$ and platform utilities: under hardmax, $\sum_i U_i(f) = V(f)$, whereas under softmax $\sum_i U_i^{\text{soft}}(f) \le V(f)$, typically with strict inequality, which further increases the misalignment between platform incentives and user welfare. Nevertheless, all comparisons between $V(f)$ and $W_{\text{opt}}$, including Lemma 4.2 and its Proposition 4.3, remain valid.
>
>
> > On the Assumption of Complete Information (W1, Q2)
>
>
> The current model indeed assumes that platforms and providers know the user-type distribution $\pi(\theta)$ and the model performance score $S_i(\theta)$. This is a standard assumption in market-design and platform-competition models [3,4], as it allows us to isolate the core strategic incentives. That said, we agree that in practice these quantities are not perfectly known. A more realistic view is that $\pi(\theta)$ and $S_i(\theta)$ are estimable primitives, which may be inferred from usage logs, public benchmarks, online learning procedures and public surveys.  For example, a provider may deploy surveys to infer the prevalence of different user types or recruit users from target groups to interact intentionally with competing models and report feedback; such data can then be used to estimate the preference scores $S_i(\theta)$.
>
> In settings with partial or asymmetric information, the relevant equilibrium concepts would naturally shift toward Bayesian Nash equilibrium or learning-in-games dynamics. Platforms would need to learn both $\pi(\theta)$ and $S_i(\theta)$ while competing, which introduces exploration-exploitation trade-offs. A full analysis of this learning-and-competition problem is a promising direction for future work. Our current framework serves as a complete-information benchmark on which such extensions can be developed.
>
> [1] Meena Jagadeesan, Michael Jordan, Jacob Steinhardt, and Nika Haghtalab. Improved bayes risk can yield reduced social welfare under competition. In Thirty-seventh Conference on Neural Information Processing Systems, 2023.
>
> [2] Meena Jagadeesan, Nikhil Garg, and Jacob Steinhardt. Supply-side equilibria in recommender systems. In Advances in Neural Information Processing Systems, 2023.
>
> [3] Yishay Mansour, Aleksandrs Slivkins, and Zhiwei Steven Wu. Competing bandits: Learning under competition. In 9th Innovations in Theoretical Computer Science Conference, 2018.
>
> [4] Jiri Hron, Karl Krauth, Michael Jordan, Niki Kilbertus, and Sarah Dean. Modeling content creator incentives on algorithm-curated platforms. In The Eleventh International Conference on Learning Representations, 2023.

---

> > ### Author Response · Authors · 2025-11-23
> >
> > > On Non-Strategic Provider and the Static Nature of Competition (W2, Q3)
> >
> > In our paper, the set of available models $G$ is treated as fixed when we analyze platform competition (Sections 3-4). Platform strategies and the resulting Nash equilibria are defined with respect to this fixed model set. Section 5 (Section 6 in the revised paper) then adds the perspective of a model provider and asks: If a new provider enters with a new model, how should they design it, **given that platforms will later react to it**?
> >
> > The key point is that the provider's objective in Section 5 (Section 6 in the revised paper) is already defined in terms of the market equilibrium that results after platforms best respond. So even though platforms are not modeled as fully dynamic players, the provider does anticipate that platforms will adjust their choices, and optimizes the model with those equilibrium responses in mind.
> >
> > Because the analysis is based on static Nash equilibrium, where all platforms choose simultaneously, the entry strategy in Section 5 is consistent with that same equilibrium concept. The entrant's payoff is computed after the platforms reach their new Nash equilibrium, so the provider is effectively optimizing with respect to that outcome.
> >
> > A richer model would allow providers and platforms to anticipate each other’s reactions across multiple rounds (e.g., strategic retraining). This multi-stage setting is much more complex and is an interesting direction for the future research. Our current framework provides a starting point that future dynamic extensions can build on.

---

### Official Review · Reviewer_1m6H · 2025-11-01

**Soundness:** 3
**Presentation:** 4
**Contribution:** 3
**Rating:** 6
**Confidence:** 5

**Summary:**

The paper presents a three-layer framework consisting of the model, platform, and users to study competition in the generative model ecosystem. In this setting, foundation model providers such as Anthropic and OpenAI license their models to platforms like Microsoft Azure and Amazon Bedrock, which then compete to attract users. The authors derive conditions under which a pure Nash equilibrium exists and describe its structure, examining whether platforms use same models (a homogeneous equilibrium) or different ones (a heterogeneous equilibrium). They then analyze user welfare and show that greater competition, either through more models or more platforms, does not always lead to higher social welfare Finally, the paper considers the perspective of model providers and explores training strategies that can improve their overall utility

**Strengths:**

The main strengths are:
1) Originality in model formulation: The paper introduces a three-layer model, platform, user framework to study competition in generative AI markets. Prior work, typically focuses on two-layer settings involving only users and platforms/models. Their framework captures the distinct incentives at each layer and the competitive interactions (among platforms and among model providers).

2) Section 3 is a highlight of the paper. It provides a clear and rigorous characterization of equilibrium conditions and market structure. The decomposition of platform utility into attraction and deviation components show when platforms converge on a single model versus when they diversify across different ones.

3) Section 5 is also notably original, introducing a strategic perspective on model training. It shows how model providers can adjust their training objectives to improve adoption by competing platforms, with the direct-gradient optimization approach standing out as an innovative method.

**Weaknesses:**

1. Platforms Limited to a Single Model:

The modelling assumes that each platform selects one model provider. This does not reflect the papers motivation where platforms like Azure and Bedrock host multiple foundation models simultaneously. As a result, the framework cannot capture strategies such as model bundling, which are important for platform differentiation and for covering diverse user needs. This is not a major weakness, but it would be useful if the authors could discuss how multiple-model selection could be incorporated into the current framework.

2. Inconclusive Welfare Insights (Section 4);

The findings in Section 4 are somewhat unclear. Figure 3 illustrates that adding a new model can reduce user welfare, highlighting the “paradox of competition.” At the same time, the authors provide sufficient conditions under which welfare increases. However, it is not clear how frequently these conditions are met in practice, and in the experiments (Section 6) user welfare appears to increase as more models are added. This makes it difficult to draw a consistent, general conclusion about the effect of competition on welfare.

**Questions:**

- Please address the question raised in the weaknesses, particularly regarding platform model choice. For example, how could the framework be extended to allow platforms to deploy multiple models, and how would this impact the analysis and equilibrium outcomes?

---

> ### Author Response · Authors · 2025-11-23
>
> We thank the reviewer for the positive and thoughtful evaluation. We are encouraged that the three-layer model and theoretical analysis are considered both original and clear. We address the two main concerns below.
>
> > Platforms Limited to a Single Model (W1, Q)
>
> We thank the reviewer for this insightful observation. Our current framework assumes that each platform selects a single model for simplicity and analytical tractability, which allows us to focus on the fundamental competitive dynamics and resulting equilibrium structure. In fact, we can extend the framework naturally to accommodate multiple models per platform. In this setting, each platform's strategy would be a set of models $M_{i}$, and user choice could depend on the highest-performing model in that set for their type, $\hat{S}\_{i}(\theta)= \max_{j \in M_{i}}S_j(\theta) $, or an expected score $\hat{S}\_{i}(\theta)= \mathbb{E}_{j \in M\_{i}}S_j(\theta)$. The three-layer market formalization remains as before, with platform payoffs computed using $\hat{S}_i(\theta)$ instead of $S_i(\theta)$, and best responses now taken over model mixtures rather than single models.
>
> Many of the existing analysis, such as the utility decomposition, equilibrium characterization, and welfare analysis, can be generalized to this setting. For example, the deviation terms can be redefined to account for the combined effect of multiple models, and the coverage-based welfare metric would reflect the union of user types served by the platform's model set. While this extension may somewhat reduce the contrast between homogeneous and fully differentiated equilibria, the core trade-off remains: platforms must balance focusing on majority-preferred models with allocating capacity to models that better serve minority user types.
>
> We have added the above discussion to the revised paper in Appendix B.
>
>
> > Inconclusive Welfare Insights (W2)
>
> Section 4 highlights that the effect of adding new models or platforms on user welfare is **non-monotone and context-dependent**, rather than to assert a universal trend. Examples in Figure 3 (Figure 4 in the revised paper) are constructed to illustrate the "paradox of competition", where additional models can reduce welfare, while the propositions in Section 4 provide sufficient conditions that guarantee welfare improvements.
>
> The frequency with which these conditions are met depends on the distribution of user types and the heterogeneity of model performance. In Figure 4 (Figure 5 in the revised paper), user welfare tends to increase with additional models because the experimental setup corresponds to scenarios where the sufficient conditions are satisfied. This, however, does not contradict the theoretical insight: as illustrated in Figure 5 (Figure 6 in the revised paper) and Example C.5, adding models can indeed reduce welfare under other distributions or heterogeneity patterns.

---

> > ### Comment · Reviewer_1m6H · 2025-11-28
> >
> > Thank you for the clarifications, I recommend that the paper be accepted

---

### Official Review · Reviewer_vtm9 · 2025-11-02

**Soundness:** 3
**Presentation:** 3
**Contribution:** 2
**Rating:** 4
**Confidence:** 4

**Summary:**

This paper formalizes GenAI ecosystems as a three-layer market game consisting of model providers, platforms, and user populations. Platforms strategically select genAI models from providers to serve user groups, while users choose platforms based on preferences. The work discusses the structure of equilibria and analyzes conditions for PNE, market differentiation, and user welfare, and provide algorithm for model provider for effective market entry.

**Strengths:**

The topic is timely and relevant, theoretical analysis is nice and solid. Presentation is easy to follow. Despite the weaknesses I pointed out below, I really like the angle from which the paper formulates the problem and the style of presentation.

**Weaknesses:**

As a theoretical-oriented paper with the claimed strength lying on the proposed game-theoretic model, my main concern is that the model structure and theoretical finds are not sufficiently interesting for providing new insights. Here is some of my thoughts:
1. The model seem to me is too stylistic. The hardmax user choice model is overly simplified, it would be more interesting to consider softmax or other alternative stochastic choice model and see if similar observation holds. If hardmax is a necessary simplification to derive theoretical results, I believe some discussion or simulations on alternative user behavior models will strengthen the paper.

2. I do not see a main theoretical claim (there are lots of proposition, lemmas and corollaries, but no theorems). And those results seems to be straightforward and lacks insights. For example,
 - non-existence of PNE is standard as in my understanding these types of games can only be shown to have PNEs if it has a concave or potential structure. I would expect a deeper understanding like if the PNE does not exist, what would a typical best-response loop look like and how it reflects some phenomenon in reality.
- Corollary 3.5 conveys a very intuitive message but if unfortunately it rely on a seemingly too restrictive set of assumption. I'm wondering how rare the situation happens when a model has a clear advantage over all other models for the majority user group, while performs similar to other models for all remaining user groups.
- Lemma 4.2 does not contain any quantitative result. Isn't it trivial to simply saying the welfare at equilibrium can be less than the globally optimal welfare? It is just the definition of the price of Anarchy.
3. The three-layer model does not seem to be tied to the nature of GenAI. The exact same model can be used to study the content creation market as well, if we view the model providers as content creator and candidate models as potential topics or genres of content creation strategies. That said, the proposed model is claimed to capture the market of genAI models but does not actually capture any nuance of the power of GenAI nor how it adds anything special to the market. I might have missed something, if so, I'd appreciate it if the author reemphasize how the competition model is uniquely relevant to genAI.
4. Section 5 provides little conceptual contribution. I'm not sure how practical the optimization procedure can be actually adopted by a GenAI model provider to determine the new market entry strategy, as it rely on transparent information of all the factors in the market. And one thing it fails to consider is the cost, which should be a very important factor for new players.
5. Limitations in experiments. CIFAR-10 setting is somewhat toy-level and I believe results on language or multimodal models would better reflect generative model ecosystems. Results are lack of statistical significance as well, no error bar seen in figures. And also investigation on the sensitivity of results to various user distributions would be highly appreciated.

Minor issues:
1. typo in Definition 2.1 (Nash Equilibirum) -> Nash Equilibrium

**Questions:**

Please refer to my questions raised in the weaknesses section.

---

> ### Author Response · Authors · 2025-11-23
>
> We thank the reviewer for the constructive feedback and are glad that our work is recognized as timely, theoretically solid, and clearly presented. Below, we address the main concerns raised.
>
> > On hardmax user choice model (W1)
>
> We thank the reviewer for this suggestion. Indeed, the hardmax user choice model is a standard simplification commonly used in prior work on platform or model selection [1-3], where users deterministically select the platform with the highest score. To address the concern, we have extended our analysis to a softmax choice model and added a section (Section 5) in the revised paper. In this model, users select platforms probabilistically according to their scores (Eq.8), with a temperature parameter $\tau$ controlling the degree of stochasticity; the hardmax model is recovered as a special case when $\tau \to 0$. We clarify in Section 5 that:
>
> - The negative result on the existence of equilibrium (Proposition 2.2) is robust to the softmax extension. Any instance with no pure Nash equilibrium (PNE) under hardmax remains without a PNE for sufficiently small $\tau$ (Proposition C.7), and there exist instances with fixed $\tau>0$ where the softmax model also admits no PNE (Example C.8).
>
> - The utility decomposition (Proposition 3.3) and the equilibrium characterization for fully differentiated and homogeneous markets (Lemma 3.4) extend directly to the softmax case once the deviation term is redefined (Definition C.9). The equilibrium conditions retain the same form and can still be expressed as inequalities involving $T_{f_i}$ and $\delta^{\text{soft}}_{f_i}(f)$. This shows that our qualitative insights, such as the role of model performance heterogeneity in supporting differentiated equilibria and the misalignment between platform incentives and user welfare, hold beyond the hardmax assumption.
>
> - For welfare analysis, the coverage-based measure $V(f)$ depends only on which models are available and is therefore unaffected by the choice of hardmax or softmax. The only change is in the relation between coverage $V(f)$ and platform utilities: under hardmax, $\sum_i U_i(f) = V(f)$, whereas under softmax $\sum_i U_i^{\text{soft}}(f) \le V(f)$, typically with strict inequality, which further increases the misalignment between platform incentives and user welfare. Nevertheless, all comparisons between $V(f)$ and $W_{\text{opt}}$, including Lemma 4.2 and its Proposition 4.3, remain valid.
>
> [1] Meena Jagadeesan, Michael Jordan, Jacob Steinhardt, and Nika Haghtalab. Improved bayes risk can yield reduced social welfare under competition. In Thirty-seventh Conference on Neural Information Processing Systems, 2023.
>
> [2] Meena Jagadeesan, Nikhil Garg, and Jacob Steinhardt. Supply-side equilibria in recommender systems. In Advances in Neural Information Processing Systems, 2023.
>
> [3] Yishay Mansour, Aleksandrs Slivkins, and Zhiwei Steven Wu. Competing bandits: Learning under competition. In 9th Innovations in Theoretical Computer Science Conference, 2018.

---

> > ### Author Response · Authors · 2025-11-23
> >
> > > Insights of our results (W2)
> >
> > We thank the reviewer for these comments and the opportunity to clarify the theoretical insights of our work.
> >
> > 1. **Insights on (Non)-existence of equilibrium**. In the newly added Section 5, we extend the hardmax model to a softmax user choice model and show that the non-existence of pure Nash equilibria (PNE) is robust to small stochastic noise: strictly no-PNE instances under hardmax remain without pure equilibria for small $\tau$. To illustrate this, we provide concrete examples (Example C.6 and Figure 4(c) (Figure 5(c) in the revised paper) for hardmax choice; Example C.8 for softmax choice) showing that best responses cycle indefinitely rather than converge. In these cases, heterogeneous user rankings over models, combined with platforms optimizing only for their own users, lead platforms to "chase" each other's majority segments and results in repeated cycles between homogenization and niche differentiation.
> >
> >     In Section 3, Proposition 3.3 and Lemma 3.4 do more than establish the existence of equilibria. By separating average model performance $T_f$ from the deviation advantage $\delta_f$ generated by heterogeneous user preferences and competitors' strategies, these results identify the precise conditions under which market segmentation occurs. Differentiated equilibria appear only when the local $\delta_f$ terms satisfy specific inequalities, i.e., when niche advantages are strong enough to overcome average performance gaps. This formalizes that segmentation is not automatic but emerges from concrete structural conditions in both user preferences and platform strategies.
> >
> > 2. **Condition for Corollary 3.5**. Corollary 3.5 illustrates how highly concentrated demand can lead to homogenization even when multiple models are available. We clarity that the only assumption needed for Corollary 3.5 to hold is hardmax user choice. To assess how rare the situation in Corollary 3.5 (i.e., $\pi_{\theta}^{\star} \ \ge\ 1-\frac{1}{1+2\frac{\Gamma}{\rho}}$) is, we plot $\pi_{\theta}^{\star}$ (the fraction of the dominant user group) as a function of  $\frac{\Gamma}{\rho}$ (the relative strength of minority variation to majority advantage) in Figure 3. The shaded region in $(\frac{\Gamma}{\rho}), \pi_{\theta}^{\star}$ space indicates where the condition holds and homogenization is inevitable. The plot shows that homogenization is not rare: for instance, if $\frac{\Gamma}{\rho}=0.5$, homogenization occurs whenever the dominant user group constitutes at least half of the population ($\pi_{\theta}^{\star}\geq 0.5$).
> >
> > 3. **Insights from the welfare analysis.** Section 4 highlights a counterintuitive effect: expanding the model pool or adding more platforms does not necessarily improve user welfare. In our framework, additional models or platforms can shift best responses in ways that reduce coverage and amplify the misalignment between platform incentives and social welfare. This demonstrates a non-trivial phenomenon and provides an important design insight for platform markets: simply increasing competition or model variety does not guarantee better outcomes for users.

---

> > > ### Author Response · Authors · 2025-11-23
> > >
> > > > The three-layer model for GenAI market (W3)
> > >
> > > We emphasize the GenAI market to distinguish it from discriminative settings studied in prior literature. As highlighted in lines 36-43 of the Introduction, existing studies on machine learning markets have largely focused on two-layered markets of supervised (discriminative) models, specifically binary classification, where a set of classifiers is available on the market and users select those with correct predictions. Compared to classifier markets, generative-model markets exhibit key differences:
> > >
> > > 1. In binary classification, each model's output is either correct or incorrect, so the possible interactions between two classifiers are tightly constrained. Outcomes can be divided into four cases: both correct, only the first correct, only the second correct, or both wrong. This gives rise to a simple "conservation rule": the difference in overall accuracy between two classifiers is fully determined by the fractions of cases where only one is correct. For example, suppose classifiers A and B are evaluated on 100 examples: both correct on 60, only A correct on 20, only B correct on 10, and both wrong on 10. Then Accuracy(A) = 0.8, Accuracy(B) = 0.7, and the difference 0.1 equals 0.2-0.1, i.e., the fraction only A is correct minus the fraction only B is correct. This rigid structure strongly constrains user preferences and model rankings, which is why two-layer classifier markets are relatively simple to analyze.
> > >
> > >     Generative models, by contrast, produce open-ended, multi-dimensional outputs that are not simply right or wrong, but vary along aspects such as fluency, style, creativity, and safety. When two generative models produce different outputs, there is no analogous conservation constraint, and user groups can naturally rank the same pair of models differently. This lack of a one-dimensional constraint makes the modeling and conclusions in GenAI market fundamentally different from those in classical classifier markets.
> > >
> > > 2. Much of the existing work effectively studies a two-layer "model-user" market, where the same player both trains and deploys a classifier. By contrast, GenAI ecosystems are naturally three-layer: model providers, platforms, and users. Platforms act as intermediaries that choose, combine, and adapt models for their own user bases, which is exactly what our model-platform-user framework captures.
> > >
> > > While motivated by GenAI, we agree with the reviewer that our framework is not limited to GenAI market. In fact, we can extend it even more broadly to (digital-)content markets as creator-produced content are also open-ended: model-providers generate contents (e.g., books, videos, music), platforms (e.g., bookstores, streaming services) select which providers to feature; and users choose platforms to interact with. In this way, our framework generalizes beyond GenAI to capture the strategic dynamics of intermediated digital markets.

---

> > > > ### Author Response · Authors · 2025-11-23
> > > >
> > > > > Contribution of Section 5 (W4)
> > > >
> > > > Section 5 (Section 6 in the revised paper) is intended to provide conceptual and illustrative guidance on how a model provider might reason about strategies to enter a three-layer market. Although the optimization framework requires knowledge of several market quantities, many of these, including user-type distributions and model-performance scores, can in practice be reliably estimated through usage logs, public benchmarks, online learning procedures, or public surveys. For example, the provider may launch surveys to infer user types and their prevalence, or recruit users of different types to intentionally interact with competing models and report feedback, which can then be used to estimate the performance scores $S_i(\theta)$.
> > > >
> > > > We agree that entry costs are an important consideration. Our framework can naturally incorporate costs by defining the provider's optimization as $ \max F(\phi):= \sum_{\theta \in \Theta} \pi_{\theta} \sigma_{\theta} S_{\phi}(\theta) - C(\phi)$, or equivalently, $ \max F(\phi):= \sum_{\theta \in \Theta} \pi_{\theta} \sigma_{\theta} S_{\phi}(\theta)$ subject to $C(\phi) < Budget$, where $C(\phi)$ represents the cost of deploying model $\phi$. Importantly, introducing costs affects only the entry optimization criterion, and **does not change the equilibrium structure or qualitative insights** analyzed in the earlier sections.
> > > >
> > > > > Experiments (W5)
> > > >
> > > > Our theoretical framework and results depend on generative models only through the scores $S_i(\theta)$, regardless of the complexity or modality of the data or the model. We use CIFAR-10 because it allows us to implement experiments efficiently while clearly illustrating the theory. The results naturally extend to richer generative model ecosystems, including text, audio, language, or multimodal models. To support this claim, we **add an additional experiment with language models in Appendix F, and observe the similar results.**
> > > >
> > > > For the sensitivity to user-type distributions, we indeed have the results with different user pool as shown in Figure 7(c),(f),(g),(h) of Synthetic Dataset and Table 7 of Real Dataset. We show the new results in Figure 12(c) and (f) of language models.
> > > >
> > > > We appreciate the reviewer's suggestion of error bars. The revised manuscript now includes error bars computed over three independent runs.

---

### Author Response · Authors · 2025-11-30
**Rebuttal Summary**

We thank all reviewers for their constructive feedback, and we are grateful that they found our three-layer model-platform-user framework both **timely and original**, the theoretical analysis **solid and clear**, and the motivation around generative-model markets **compelling**. Below we summarize the main concerns raised and how we addressed them during the rebuttal phase:

1. **From hardmax to softmax user choice:** Reviewers questioned the realism of the hardmax user-choice rule and asked how the analysis would change under more realistic stochastic choice models (e.g., softmax) (reviewer `vtm9` and `ZFbA`). In the revision, we add a softmax user choice model in Section 5 and show that the non-existence of PNE, the equilibrium structure, and our coverage-based welfare comparisons all remain robust, while platform utilities become even more misaligned with welfare. This demonstrates that our structural and welfare insights persist under relaxed user-choice assumptions.

2. **Multi-task services and multi-model platforms:** Reviewers raised concerns about our use of a single reward (reviewer `1m6H`) and the assumption that each platform deploys only one model (reviewer `c5YV`). We clarify that $S_i(\theta)$ is a type-dependent score that can aggregate multiple quality aspects differently for different user groups, so the framework already captures heterogeneous, multi-task preferences within a shared model pool. In the revision, we add a discussion in Appendix B of platforms that offer multiple services and thus share only a subset of the model pool, noting that such partially overlapping markets are a promising direction for future work. We also explain how the framework can be extended to allow each platform to deploy a set of models, with user choice based on the best or the expected score from that set.

3. **Information assumptions:** Reviewers raised concerns about the complete-information assumption that platforms and providers know the user-type distribution $\pi(\theta)$ and the performance scores $S_i(\theta)$ (reviewer `vtm9` and `ZFbA`). We clarify that this assumption is standard in market-competition models, and that in practice these quantities can be viewed as estimable primitives that can be inferred from usage logs, public benchmarks, and surveys. We also briefly discuss that, under partial or asymmetric information, the relevant concepts shift to Bayesian Nash equilibria or learning-in-games dynamics, and we highlight this learning-and-competition setting as an important direction for future work, with the current model serving as a clean complete-information baseline.

4. **Dynamics and multi-round behavior:** Several comments asked how our static analysis relates to multi-round, stateful dynamics (reviewer `c5YV` and `ZFbA`). We clarified that:
- Best-response dynamics in the paper are used as an analytical tool to study a static game and to test for existence of pure Nash equilibria (including explicit examples that generate cycles), not as a literal model of how users and platforms update in real time. Our main results are stated in terms of static Nash equilibria.
- For model entry, the provider's optimization problem is already defined with respect to the post-entry equilibrium: the provider optimizes a model whose payoff is computed after platforms best respond and reach the new Nash equilibrium. In that sense, the entry strategy is consistent with the same equilibrium concept used in our platform analysis and does not assume platforms remain frozen.
- We acknowledge that capturing such dynamics would require a significantly more complex framework (for example, performative prediction in a stateful world), and we highlight this in the new Appendix B as a natural follow-up direction building on our three-layer first-round baseline.

5. **Experiments** (reviewer `vtm9`): We add error bars, clarify the sensitivity to user-type distributions more explicitly, and include an additional language-model experiment in the appendix to support the claim that the theory applies beyond CIFAR-10.

---

> ### Author Response · Authors · 2025-11-30
>
> Based on the feedback, we have revised the paper accordingly. Below we summarize the main changes:
>
> 1. We **refined the discussion of Corollary 3.5** and added a plot of  $(\frac{\Gamma}{\rho}, \pi_{\theta}^{\star})$ to illustrate when the homogenization condition holds, addressing the reviewers' concern (reviewer `vtm9`) about how intuitive and frequent this regime is.
>
> 2. We added a new section (**Section 5**) in the main text that extends the user choice rule from hardmax to softmax. In addition, **Appendix C.10** now provides the corresponding proofs, extended definitions, and concrete examples for the softmax setting.
>
> 3. We added a new discussion (**Appendix B**) that synthesizes the reviewers' questions on complete information, platforms deploying multiple models, and multi-round or dynamic competition, and explains how our framework can be interpreted or extended in these directions.
>
> 4. We added **Appendix F** with experiments on language-model-based services and updated all experimental figures to include error bars computed over multiple runs, thereby strengthening the empirical support for our theoretical insights.
>
> Overall, we thank all reviewers again for their valuable suggestions, which have significantly improved the paper, and we are grateful to the AC for the additional time and effort spent evaluating our work.

---

### Meta-Review · Area_Chair_YzWA · 2026-01-06

**Summary:**

This paper proposes a game-theoretic model of a model-platform-user market for studying the strategic nature of the current generative AI market. The reviewers found the 3-layer model to be sound, sensible, and clearly presented. There were concerns regarding the assumptions. The assumption that users choose the platform with the highest score was challenged by several reviewers, and so the authors explored a softmax (stochastic) choice model and showed their results were robust to this change. The authors also discussed relaxing other assumptions (allowing incomplete info w/ Bayesian NE and taking average score over multi-model platforms in Appx. B), but addressing these rigorously is largely left to future work.

**Reviewer Concerns:**

I believe all reviewer concerns were addressed in the authors rebuttal as summarized including extensions to softmax user choice, multi-task services and multi-model platforms, information assumptions, dynamics and multi-round behavior, and experiments. Some of these concerns are addressed comprehensively such as the softmax user choice model and additional experiments. The authors explain that assumptions on complete information and single-model platforms can be relaxed and their theory can be generalized although this is left to be established rigorously in future work. Exploring dynamics is left to future work as well.

**Reviewer Scores:**

- vtm9: I think the reviewer might have increased their score by +1 or +2 to a 5 or 6.
- 1m6H: The reviewer responded in a comment that they recommend the paper be accepted to I expect they would have raised their score to a 8.
- ZFbA: I expect the reviewer would have increased their score by +1 to a 7 in light of the new softmax results.
- c5YV: I think they would keep their score.

---

### Decision · Program_Chairs · 2026-01-26

Accept (Poster)